# Socio-demographic factors associated with normal linear growth among pre-school children living in better-off households: A multi-country analysis of nationally representative data

Dickson Abanimi Amugsi[1]*, Zacharie T. Dimbuene[2,3], Elizabeth W. Kimani-Murage[1]

**1** Maternal and Child Wellbeing Unit, Research Division, African Population and Health Research Center, Nairobi, Kenya, **2** Department of Population Sciences and Development, University of Kinshasa, Kinshasa, Democratic Republic of the Congo, **3** Microdata Access Division, Statistics Canada, Ottawa, Canada

☯ These authors contributed equally to this work.
\* damugsi2002@yahoo.com, damugsi@aphrc.org

**Data Availability Statement:** The data underlying the results presented in the study are available from The DHS Program at http://dhsprogram.com/

## Abstract

This study examined the socio-demographic factors associated with normal linear growth among pre-school children living in better-off households, using survey data from Ghana, Kenya, Nigeria, Mozambique and Democratic Republic of Congo (DRC). The primary outcome variable was child height-for-age z-scores (HAZ), categorised into HAZ≥-2SD (normal growth/not stunted) and HAZ<-2 (stunted). Using logistic regression, we estimated adjusted odds ratios (aORs) of the factors associated with normal growth. Higher maternal weight (measured by body mass index) was associated with increased odds of normal growth in Mozambique, DRC, Kenya and Nigeria. A unit increase in maternal years of education was associated with increased odds in normal growth in DRC (aOR = 1.06, 95% CI = 1.03, 1.09), Ghana (aOR = 1.08, 95% CI = 1.04, 1.12), Mozambique (aOR = 1.08, 95% CI = 1.05, 1.11) and Nigeria (aOR = 1.07, 95% CI = 1.06, 1.08). A year increase in maternal age was positively associated with normal growth in all the five countries. Breastfeeding was associated with increased odds of normal growth in Nigeria (aOR = 1.30, 95% CI = 1.16, 1.46) and Kenya (aOR = 1.37, 95% CI = 1.05, 1.79). Children of working mothers had 25% (aOR = 0.75, 95% CI = 0.60, 0.93) reduced odds of normal growth in DRC. A unit change in maternal parity was associated with 10% (aOR = 0.90, 95% CI = 0.84, 0.97), 23% (aOR = 0.77, 95% CI = 0.63, 0.93), 25% (aOR = 0.75, 95% CI = 0.69, 0.82), 6% (aOR = 0.94, 95% CI = 0.89, 0.99) and 5% (aOR = 0.95, 95% CI = 0.92, 0.99) reduced odds of normal growth in DRC, Ghana, Kenya, Mozambique and Nigeria, respectively. A child being a male was associated with 18% (aOR = 0.82, 95% CI = 0.68, 0.98), 40% (aOR = 0.60, 95% CI = 0.40, 0.89), 37% (aOR = 0.63, 95% CI = 0.51, 0.77) and 21% (aOR = 0.79, 95% CI = 0.71, 0.87) reduced odds of normal child growth in DRC, Ghana, Kenya and Nigeria, respectively. In conclusion, maternal education, weight, age, breastfeeding and antenatal care are positively associated with normal child growth. In contrast, maternal parity, employment, and

publications/publication-fr221-dhs-final-reports.
cfm. Data are accessible free of charge upon a
registration with the Demographic and Health
Survey program (The DHS Program).

**Funding:** The authors received no specific funding
for this work

**Competing interests:** The authors have declared
that no competing interests exist

child sex and age are associated negatively with normal growth. Interventions to improve
child growth should take into account these differential effects.

## Introduction

Child health is a fundamental public health issue because good child health sets one up for life-
long health and functioning, and wellbeing. In sub-Saharan Africa (SSA), child physical health
is of particular concern due to the high rates of illness and mortality in the region. Normal
(healthy) child growth, defined in this paper as children who are not stunted (not too short for
their age), is a foundation for optimal child health and wellbeing. There is evidence that
healthy child growth is positively associated with cognitive development, higher school
achievements, lower morbidity and mortality, higher economic productivity in adulthood and
better maternal reproductive outcomes [1–3]. Thus, suggesting the need for substantial invest-
ment in nutrition interventions to promote child growth to ensure life-long benefits. Working
with international partners such as World Health Organisation (WHO) and United Nations
Children's Fund (UNICEF), governments in SSA have put in place various interventions to
improve child growth by addressing stunting in the region [4, 5]. However, the implementa-
tion of these programmes tends to focus more on child growth deficiencies and how to protect
children against risk factors of growth deficiencies [6, 7]. Therefore, it may be difficult to
directly attribute the effects of the programmes on child healthy growth outcomes, except to
infer that reduction in stunting implies an increase in healthy growth. The present study fills
this gap by providing evidence on the critical socio-demographic factors associated with
healthy growth among children living in better-off households. Indeed, many experts have
called for this type of resource-focused approach in promoting child health outcomes, as
exemplified by the UNICEF childcare framework [8, 9].

Several factors affect child linear growth in low and middle incomes countries (LMICs).
These factors include maternal education, employment, household wealth index, antenatal
care (ANC), parity, maternal body mass index (BMI), urban place of residence, breastfeeding,
and maternal age among others [10–30]. These factors affect child linear growth either nega-
tively or positively. There is substantial evidence that improvement in maternal education has
a significant positive effect on child growth outcomes in many settings [10–14]. Educated
mothers tend to have children with better nutritional status compared to children of mothers
with no education.

Similarly, maternal BMI has a strong positive effect on child linear growth [10, 11, 14–17].
A study in Ethiopia showed that maternal BMI was associated positively with children nutri-
tional status [17]. Fenske and colleagues [11] observed that maternal age has a significant effect
on childhood stunting. In India, undernutrition was more prevalent in children of 26–30 year
age group mothers than the other reproductive age groups [18]. Relatedly, children of older
mothers tend to suffer less from stunting compared to children of younger mothers [12, 19,
20]. Although there is scanty literature on the effects of maternal parity on child nutritional
status, some few studies have observed negative associations between maternal parity and
child growth outcomes [21–23].

Further, Kuhnt and Vollmer[24] found in their study that having at least four ANC visits is
associated with reduced odds of stunting in pre-school children. Several studies have observed
a positive effect of breastfeeding on child growth [25–27]. Household wealth index also has a

strong positive impact on child growth [28–30]. Children in better-off households tend to have better growth outcomes relative to those in poor households.

The literature reviewed above focus almost exclusively on child growth deficiencies (under-nutrition/abnormal growth) and the associated risk factors. Statistical analyses that examine the direct relationship between socio-demographic factors and healthy child growth are still limited, to the best of our knowledge. Therefore, it is significant to conduct analysis, using healthy growth as the primary outcome variable to elucidate the direct relationship between socio-demographic factors and healthy linear growth among children. Furthermore, it is widely recognised that children in better-off households tend to have better growth and health outcomes. However, stratified analysis to understand the key covariates responsible for the positive growth outcomes in this sub-group is lacking. This study intended to fill this gap by stratifying the analysis by better-off households, focusing on children who are growing normally (rather than those that are not) and the factors that make them grow well. We also investigated the factors that pose potential risks to child growth in better-off households. This investigation will further our understanding of the critical factors associated with healthy child growth. The objective of this study, therefore, was to examine the associations between socio-demographic factors at child, maternal, household and community levels and healthy growth among children living in better-off households.

## Methodology

**Data sources and sampling strategy.** This analysis used data from the Demographic and Health Surveys (DHS) [31], conducted in Ghana (2014), Kenya (2014), Nigeria (2013), Mozambique (2011) and Democratic Republic of Congo (DRC) (2013–2014). We based the selection of the five countries on our previous analysis using the same countries and data [12, 32]. The DHS data are nationally representative, repeated cross-sectional household surveys collected primarily in LMICs every five years using standardised questionnaires to enable cross-country comparisons [33, 34]. The DHS utilises a two-stage sample design [35–39]. The first stage involves the selection of sample points or clusters from an updated master sampling frame constructed from National Population and Housing Census of the respective countries [37]. DHS selected the clusters using systematic sampling with probability proportional to size. Household listing is then conducted in all the selected clusters to provide a sampling frame for the second stage selection of households [12, 37]. The second stage selection involves the systematic sampling of the households listed in each cluster and randomly select from the list the households to be included in the survey [12, 37]. The rationale for the second stage selection is to ensure adequate numbers of completed individual interviews to provide estimates for critical indicators with acceptable precision. All men and women aged 15–59 and 15–49 respectively, in the selected households (men in half of the households) are eligible to participate in the surveys if they were either usual residents of the household or visitors present in the home on the night before the study [12, 37].

**Study participants.** The study population comprised children aged 0–59 months, born to mothers aged 15–49 years living in better-off households. The DHS obtained information on the children through face-to-face interviews with their mothers. Adjustable measuring board calibrated in millimetres was used in measuring study children height. Children younger than 24 months were measured lying down (recumbent length) on the board while older children were measured standing [12]. The DHS then converted the height data into Z-scores based on the 2006 WHO growth standards [40]. The total samples used in the current analysis were: Ghana, n = 1,247; Nigeria, n = 12,999; Kenya, n = 3,895; Mozambique, n = 5,711; and DRC, n = 3,943.

## Ethics statement

Before conducting the surveys, the DHS obtained ethical clearance from Government recognised Ethical Review Committees/Institutional Review Boards of the respective countries as well as the Institutional Review Board of ICF International, USA. Study children mothers or caregivers gave written informed consent before the inclusion of their children. The authors of this paper sought and obtained permission from the DHS program for the use of the data. The data were utterly anonymised, and therefore, the authors did not seek further ethical clearance before their use.

## Outcome and predictor variables

**Outcome variables.** The primary indicator of child linear growth used in this analysis was child height-for-age Z-scores (HAZ). The HAZ scores were computed using 2006 WHO growth standards [40] and classified into healthy growth (or not stunted) and stunted (or poor growth). In this paper, we described children who have HAZ equal to or above -2 SD (HAZ $\geq$ -2SD) [40, 41] as having a healthy growth. In contrast, children with HAZ below –2 SD (HAZ $<$ -2) from the median HAZ of the WHO reference population[40] were considered stunted (chronically malnourished).

**Predictor variables.** We classified the predictor variables into child (dietary diversity, age and sex), maternal (body mass index, education, age, work status, parity, breastfeeding status, marital status, antenatal attendance), household (sex of household head, household size, number of children under 5 years) and community (place of residence) level variables. The child dietary diversity (DD) was created by counting the number of food groups the mother reported the child had consumed in the past 24h before the interview. Following recommended procedures in the construction of the child DD indicator [42], we regrouped the food types in the data into seven [7] main categories [12]: (i) grains, roots and tubers; (ii) legumes and nuts; (iii) dairy products; (iv) flesh foods and organ meats; (v) vitamin A-rich fruits and vegetables; (vi) eggs; and (vii) other fruits and vegetables. A value of 1 was given for the child's consumption of any of the food groups within 24h, while we assigned 0 for non-consumption [12]. These scores were then summed up to obtain the DD score, ranging from 0 to 7, and used in the analysis as a categorical predictor variable. Also, we assigned a score of 1 to a mother who indicated she was still breastfeeding, and 0 for "no" response. Marital status was recoded into "not in union" "cohabiting" and "married". Maternal body mass index (BMI), also referred to as Quetelet's Index [43], was derived by dividing weight in kilograms by the squared height in meters [12]. The BMI (kg/m$^2$) was then classified into BMI$<$18.50 kg/m$^2$ (underweight), BMI = 18.50–24.99 kg/m$^2$ (normal weight), BMI = 25.0–29.9 kg/m$^2$ (overweight) and BMI$\geq$ 30.0 kg/m$^2$ (obese) [44]. We used BMI$<$18.50 kg/m$^2$ (underweight) as a referenced category in the analysis. We selected the predictor variables based on the UNICEF conceptual framework of childcare [9] and the literature. We subjected the selected variables to bivariate analysis to establish their relationship with the outcome variable. Only statistically significant variables were included in the multivariable analysis. We tested statistical significant associations of the categorical variables using chi-square, while continuous variables were tested using t-test. Additionally, we included variables that were not statistically significant but considered biologically important in the multivariable analysis.

**Stratification variable.** We used the household wealth index (WI) as the stratification variable in the analysis. Several DHS reports used the WI to estimate inequalities in household characteristics, in the use of health and other services, and health outcomes [34, 35, 37, 45]. It is an indicator of wealth that is consistent with expenditure and income measurement among households [33, 34, 37]. The index was created based on assets ownership and housing characteristics of each household: type of roofing, and flooring material, source of drinking water,

sanitation facilities, ownership of television, bicycle, motorcycle, automobile among others. A principal component analysis was employed to assign weights to each asset in each household. The asset scores were summed up, and individuals ranked according to the household score. The DHS then divided the WI into quintiles: poorest, poorer, middle, richer and richest [33, 34, 37]. For this analysis, we recoded middle, richer and richest households into "better-off households". We restricted all the investigations to this sub-category.

**Framework underpinning the analysis.** The UNICEF conceptual framework [46], which outlines the causes of undernutrition underpinned our empirical analysis. It is a socio-ecological model encompassing factors at the individual, household and societal levels. In the UNICEF framework, child malnutrition is analysed in terms of immediate, underlying and basic causes. The immediate causes are inadequate dietary intakes and infectious disease, the underlying causes are inadequate maternal, and childcare, inadequate health services and healthy environment and the basic causes are institutional and socio-economic determinants and potential resources [46]. However, the extended UNICEF model of care guided this analysis [9, 46]. This framework suggests that child survival, growth and development are influenced by a web of factors, with three underlying determinants being food security, healthcare and a healthy environment, and care for children and women [9]. Basic determinants have a direct influence on these underlying determinants. These basic determinants may be described as "exogenous" determinants, which influence child nutrition through their effect on the intervening proximate determinants (underlying determinants). The underlying factors are, therefore, endogenously determined by the exogenous factors [46]. In this analysis, we included only the basic factors (socio-demographic factors) in our empirical models. We did this because there is evidence that in examining the association between child growth outcomes and exogenous factors, the proximate factors (endogenous factors) are usually excluded to avoid biased estimation of the regression parameters of the exogenous factors [47–49]. It is the case because the proximate factors are pathways through which the exogenous factors influence child nutrition [48]. Besides the basic factors, we also included antenatal care (ANC) and breastfeeding practices, which relies mostly on exogenous public health provisions rather than socio-demographic endowments of the household [46]. We included the two variables in the models' because changes in them are likely to be more responsive to policies, programmes and interventions rather than to differences in socio-demographic endowments of the household [46]. For example, there is evidence that policy, institutional and contextual settings are critical determinants of the prevalence of breastfeeding practices [47, 50].

## Data analysis

We built two empirical regression models for each of the five countries. In the first models, we included maternal BMI, education, age, work status, parity, breastfeeding status, marital status, antenatal attendance, sex of household head, household size, number of children under five years and place of residence. We adjusted for child DD, age and sex in the second and final model. We estimated adjusted odds ratios (aORs) of the associations between the socio-demographic factors and healthy growth among children in better-off households. Because the DHS utilised complex sample design, we accounted for design effect in all the analysis using parameters such as primary sampling unit, strata and weight.

## Results

### Characteristics of study samples

The results showed that Ghana (87%) had the highest number of children with healthy growth followed by Kenya (80%), while in Mozambique, DRC and Nigeria, the prevalence ranged

from 62% to 74%. Regarding dietary diversity intake, Mozambique had the highest prevalence of children who consumed at least four food groups (21%), with DRC (12%) having the lowest prevalence. Similarly, Mozambique had the highest number of women with a healthy weight (73%), followed by DRC (69%). The prevalence ranged from 43% to 59% in Ghana, Kenya and Nigeria. Ghana had the highest prevalence (59%) of women who had attained a secondary school education, while Mozambique had the lowest prevalence (21%). Higher education was less than 15% among women across all countries, with Mozambique (1.3%) registering the lowest prevalence. DRC had the highest prevalence (86%) of ANC attendance among women followed by Ghana (72%), while Nigeria had the lowest prevalence (19%). Also, Mozambique (35%) had the highest number of women who were household heads, with Nigeria (14%) having the lowest women-headed households (Table 1).

## Multivariable results of the factors promoting or inhibiting healthy child growth

The results of the multivariable analysis of the associations between maternal, child, household and community-level factors, and healthy growth among children under five years in better-off households are presented in Tables 2–6.

## Factors associated positively with normal child growth

The results showed that normal maternal weight (measured by BMI) was associated with increased odds of normal linear growth among children in Mozambique (aOR = 1.82, 95% CI = 1.33, 2.50). Similarly, overweight associated with increased odds of normal linear growth in DRC (aOR = 1.75, 95% CI = 1.17,2.62), Kenya (aOR = 2.01, 95% CI = 1.28, 3.16), Mozambique (aOR = 2.19, 95% = 1.53, 3.14) and Nigeria (aOR = 1.44, 95% = 1.16,1.77) relative to children of underweight mothers. Maternal obesity had a similar effect on normal linear growth in Kenya, Mozambique and Nigeria. One year increase in maternal years of education was associated with increased odds in normal linear growth among children in DRC (aOR = 1.06, 95% CI = 1.03, 1.09), Ghana (aOR = 1.08, 95% CI = 1.04, 1.12), Mozambique (aOR = 1.08, 95% CI = 1.05, 1.11) and Nigeria (aOR = 1.07, 95% CI = 1.06, 1.08). The results in Kenya did not reach statistical significance. An additional year in maternal age was associated with increased odds of normal linear growth among children in all the countries included in the analysis. Breastfeeding was associated with increased odds of normal linear growth in Nigeria (aOR = 1.30, 95% CI = 1.16, 1.46) and Kenya (aOR = 1.37, 95% CI = 1.05, 1.79). In Ghana, Mozambique and DRC, breastfeeding was associated positively with normal linear growth in the first model. Still, this significant statistical association disappeared after the child level covariates such as age, sex and dietary diversity were included in the final empirical model. Urban place of residence was associated with increased odds of normal linear growth among children in DRC (aOR = 1.32, 95% CI = 1.06, 1.65) and Mozambique (aOR = 1.18, 95% CI = 1.01, 1.38). The association did not reach statistical significance in the remaining three countries.

## Factors associated negatively with normal child growth

This section examines the factors associated negatively with normal linear growth. The results showed that children of mothers who were working had 25% (aOR = 0.75, 95% CI = 0.60, 0.93) reduced odds of normal linear growth in DRC. A unit change in maternal parity was associated with 10% (aOR = 0.90, 95% CI = 0.84, 0.97), 23% (aOR = 0.77, 95% CI = 0.63, 0.93), 25% (aOR = 0.75, 95% CI = 0.69, 0.82), 6% (aOR = 0.94, 95% CI = 0.89, 0.99) and 5% (aOR = 0.95, 95% CI = 0.92, 0.99) reduced odds of normal linear growth among children in DRC, Ghana, Kenya, Mozambique and Nigeria respectively. Similarly, a unit change in child's

**Table 1. Characteristics of the study samples of the five countries.**

| Variables | DRC | | Ghana | | Kenya | | Mozambique | | Nigeria | |
|---|---|---|---|---|---|---|---|---|---|---|
| | %/mean | SD | %/mean | SD | %/mean | SD | %/mean | SD | %/mean | SD |
| **Child-level covariates** | | | | | | | | | | |
| Height-for-age (HAZ ≥-2) | 62.0 | | 87.0 | | 80.0 | | 66.0 | | 74.2 | |
| DD < 4 food groups | 87.6 | | 87.0 | | 80.2 | | 78.6 | | 86.9 | |
| DD > = 4 food groups | 12.4 | | 14.0 | | 19.8 | | 21.4 | | 13.1 | |
| **Gender** | | | | | | | | | | |
| Female | 50.4 | | 48.0 | | 49 | | 49.2 | | 49.5 | |
| Male | 49.6 | | 52.0 | | 51.0 | | 50.8 | | 50.5 | |
| **Mother-level covariates** | | | | | | | | | | |
| *Body Mass Index (BMI)* | | | | | | | | | | |
| BMI <18.50 | 9.4 | | 2.9 | | 5.34 | | 4.11 | | 5.42 | |
| BMI = 18.50–24.99 | 69.2 | | 43.0 | | 55.1 | | 73.9 | | 58.7 | |
| BMI = 25–29.99 | 16.3 | | 32.0 | | 27.0 | | 17.2 | | 24.6 | |
| BMI > = 30 | 3.9 | | 22.0 | | 12.6 | | 4.54 | | 10.9 | |
| **Education** | | | | | | | | | | |
| No education | 12.6 | | 15.0 | | 6.0 | | 24.5 | | 20.6 | |
| Primary education | 38.7 | | 18.0 | | 50.8 | | 53.4 | | 24.3 | |
| Secondary education | 46.8 | | 59.0 | | 30.5 | | 20.8 | | 43.6 | |
| Higher education | 1.8 | | 7.5 | | 12.6 | | 1.26 | | 11.5 | |
| **Employment status** | | | | | | | | | | |
| Not working | 27.2 | | 25.0 | | 36.4 | | 63.3 | | 25.3 | |
| Is working | 72.5 | | 74.0 | | 63.4 | | 36.7 | | 74.3 | |
| Parity | 4.37 | 2.58 | 2.99 | 1.69 | 3.127 | 2.05 | 3.51 | 2.12 | 3.90 | 2.32 |
| Is Breastfeeding (yes) | 67.8 | | 53.0 | | 51.3 | | 55.8 | | 51.5 | |
| **Marital status** | | | | | | | | | | |
| Not in union | 13.3 | | 13.0 | | 14.7 | | 17.3 | | 5.1 | |
| Married | 67.1 | | 65.0 | | 79.6 | | 45.0 | | 90.5 | |
| Cohabiting | 19.7 | | 22.0 | | 5.62 | | 37.6 | | 4.4 | |
| **Antenatal attendance (ANC)** | | | | | | | | | | |
| Number of ANC visits > = 4 | 86.3 | | 72.0 | | 50.8 | | 45.1 | | 48.2 | |
| **Household-level covariates** | | | | | | | | | | |
| *Sex of household head* | | | | | | | | | | |
| Household head is Female | 19.7 | | 29.0 | | 29.6 | | 34.5 | | 13.9 | |
| Household head is Male | 80.3 | | 71.0 | | 70.4 | | 65.5 | | 86.1 | |
| Household size | 7.30 | 2.99 | 5.0 | 1.93 | 5.57 | 2.31 | 6.49 | 2.89 | 6.73 | 3.56 |
| Number of children under 5 | 2.28 | | 1.6 | 0.69 | 1.64 | 0.76 | 1.92 | 0.94 | 2.14 | 1.11 |
| **Community-level covariates** | | | | | | | | | | |
| Urban residence | 49.2 | | 73.0 | | 52.3 | | 45.2 | | 55.1 | |

DD = Dietary diversity; DRC = Democratic Republic of Congo; SD = Standard deviation

age was associated with reduced odds of normal linear growth in DRC, Mozambique and Nigeria. Further, a child being a male was associated with 18% (aOR = 0.82, 95% CI = 0.68, 0.98), 40% (aOR = 0.60, 95% CI = 0.40, 0.89), 37% (aOR = 0.63, 95% CI = 0.51, 0.77) and 21% (aOR = 0.79, 95% CI = 0.71, 0.87) reduced odds of normal linear growth among children in DRC, Ghana, Kenya and Nigeria respectively. The association in Mozambique did not reach statistical significance.

**Table 2.  Adjusted odd ratios (aOR) of factors associated with normal growth among children living in better-off households, DRC.**

| Variables | Model 1 | Model 2 |
|---|---|---|
| **Mother-level covariates** | | |
| BMI (kg/m$^2$) = 18.50–24.99 | 1.093 | 1.124 |
| | (0.782–1.526) | (0.796–1.589) |
| BMI (kg/m$^2$) = 25–29.99 | 1.685*** | 1.751*** |
| | (1.138–2.493) | (1.172–2.617) |
| BMI (kg/m$^2$) > = 30 | 1.573 | 1.729 |
| | (0.855–2.892) | (0.884–3.382) |
| Maternal education (in single years) | 1.061*** | 1.060*** |
| | (1.032–1.090) | (1.030–1.091) |
| Age of the mother (in years) | 1.039*** | 1.051*** |
| | (1.015–1.063) | (1.026–1.077) |
| Employment status = working mother | 0.741*** | 0.747** |
| | (0.597–0.920) | (0.597–0.934) |
| Parity | 0.902*** | 0.899*** |
| | (0.844–0.965) | (0.838–0.965) |
| Is Breastfeeding = Yes | 1.678*** | 1.184 |
| | (1.370–2.055) | (0.943–1.487) |
| Marital Status = Married | 1.034 | 1.076 |
| | (0.756–1.415) | (0.786–1.473) |
| Marital Status = Cohabiting | 0.976 | 0.994 |
| | (0.687–1.389) | (0.697–1.418) |
| Number of antenatal visits = 4+ visits | 1.589*** | 1.159 |
| | (1.296–1.947) | (0.923–1.457) |
| **Household-level covariates** | | |
| Head of HH = Male | 0.865 | 0.868 |
| | (0.681–1.098) | (0.678–1.111) |
| Household size | 1.007 | 1.002 |
| | (0.968–1.048) | (0.962–1.043) |
| Number of children under 5 years | 0.945 | 0.954 |
| | (0.844–1.059) | (0.850–1.071) |
| **Community-level covariates** | | |
| Urban residence = Urban | 1.322** | 1.320** |
| | (1.066–1.640) | (1.059–1.646) |
| **Child-level covariates** | | |
| Dietary Diversity (DD) > = 4 | | 1.294* |
| | | (0.971–1.724) |
| Age of the child (in months) | | 0.975*** |
| | | (0.969–0.982) |
| Gender = Male | | 0.815** |
| | | (0.677–0.982) |
| **Observations** | **3,943** | **3,943** |

95% Confidence Intervals (CIs) in parentheses; DD-Dietary diversity; HH-Household head; BMI-Body mass index

*** p<0.01

** p<0.05

* p<0.1

**Table 3. Adjusted odd ratios (aOR) of factors associated with normal growth among children living in better-off households, Ghana.**

| Variables | Model 1 | Model 2 |
|---|---|---|
| **Mother-level covariates** | | |
| BMI (kg/m$^2$) = 18.50–24.99 | 0.692 | 0.725 |
| | (0.224–2.136) | (0.241–2.181) |
| BMI (kg/ m$^2$) = 25–29.99 | 1.230 | 1.270 |
| | (0.385–3.929) | (0.410–3.931) |
| BMI (kg/ m$^2$) > = 30 | 1.617 | 1.700 |
| | (0.478–5.469) | (0.514–5.628) |
| Maternal education (in single years) | 1.078*** | 1.081*** |
| | (1.037–1.121) | (1.039–1.124) |
| Age of the mother (in years) | 1.047* | 1.052** |
| | (0.998–1.099) | (1.001–1.105) |
| Employment status = working mother | 1.276 | 1.264 |
| | (0.821–1.983) | (0.804–1.987) |
| Parity | 0.779** | 0.766*** |
| | (0.643–0.944) | (0.630–0.932) |
| Is Breastfeeding = Yes | 1.769** | 1.748* |
| | (1.129–2.771) | (0.981–3.116) |
| Marital Status = Married | 1.956** | 2.104** |
| | (1.032–3.708) | (1.101–4.022) |
| Marital Status = Cohabiting | 1.524 | 1.659 |
| | (0.783–2.967) | (0.849–3.240) |
| Number of antenatal visits = 4+ visits | 1.124 | 1.110 |
| | (0.695–1.817) | (0.646–1.906) |
| **Household-level covariates** | | |
| Head of HH = Male | 1.250 | 1.200 |
| | (0.783–1.996) | (0.753–1.911) |
| Household size | 0.998 | 0.998 |
| | (0.894–1.114) | (0.891–1.117) |
| Number of children under 5 | 0.834 | 0.843 |
| | (0.555–1.253) | (0.559–1.274) |
| **Community-level covariate** | | |
| Urban residence = Urban | 1.020 | 0.988 |
| | (0.640–1.628) | (0.623–1.565) |
| **Child-level covariates** | | |
| Dietary Diversity (DD) > = 4 | | 1.022 |
| | | (0.590–1.770) |
| Age of the child (in months) | | 0.998 |
| | | (0.982–1.014) |
| Gender = Male | | 0.596** |
| | | (0.400–0.887) |
| **Observations** | **1,247** | **1,247** |

95% Confidence Intervals (CIs) in parentheses; DD-Dietary diversity; HH-Household head; BMI-Body mass index

*** p<0.01

** p<0.05

* p<0.1

**Table 4. Adjusted odd ratios (aOR) of factors associated with normal growth among children living in better-off households, Kenya.**

| Variables | Model 1 | Model 2 |
|---|---|---|
| **Mother-level covariates** | | |
| BMI (kg/ m$^2$) = 18.50–24.99 | 1.245 | 1.313 |
| | (0.824–1.881) | (0.872–1.976) |
| BMI (kg/ m$^2$) = 25–29.99 | 1.880*** | 2.014*** |
| | (1.195–2.960) | (1.284–3.158) |
| BMI (kg/ m$^2$) > = 30 | 1.860** | 1.991** |
| | (1.082–3.195) | (1.165–3.402) |
| Maternal education (in single years) | 1.017 | 1.021 |
| | (0.984–1.051) | (0.987–1.055) |
| Age of the mother (in years) | 1.074*** | 1.075*** |
| | (1.046–1.102) | (1.046–1.104) |
| Employment status = working mother | 0.969 | 0.982 |
| | (0.775–1.211) | (0.786–1.227) |
| Parity | 0.752*** | 0.752*** |
| | (0.688–0.823) | (0.687–0.822) |
| Is Breastfeeding = Yes | 1.373*** | 1.367** |
| | (1.103–1.710) | (1.047–1.785) |
| Marital Status = Married | 1.131 | 1.112 |
| | (0.826–1.547) | (0.810–1.526) |
| Marital Status = Cohabiting | 1.278 | 1.233 |
| | (0.785–2.079) | (0.756–2.009) |
| Number of antenatal visits = 4+ visits | 1.132 | 1.095 |
| | (0.906–1.415) | (0.862–1.390) |
| **Household-level covariates** | | |
| Head of HH = Male | 0.986 | 0.987 |
| | (0.781–1.245) | (0.780–1.248) |
| Household size | 1.054* | 1.047 |
| | (0.991–1.122) | (0.984–1.114) |
| Number of children under 5 | 1.018 | 1.026 |
| | (0.871–1.189) | (0.878–1.200) |
| **Community-level covariate** | | |
| Urban residence = Urban | 0.883 | 0.890 |
| | (0.719–1.085) | (0.723–1.096) |
| **Child-level covariates** | | |
| Dietary Diversity (DD) > = 4 | | 0.784* |
| | | (0.597–1.030) |
| Age of the child (in months) | | 0.996 |
| | | (0.989–1.003) |
| Gender = Male | | 0.625*** |
| | | (0.507–0.770) |
| **Observations** | **3,985** | **3,985** |

95% Confidence Intervals (CIs) in parentheses; DD-Dietary diversity; HH-Household head; BMI-Body mass index

*** p<0.01

** p<0.05

* p<0.1

**Table 5. Adjusted odd ratios (aOR) of factors associated with normal growth among children living in better-off households, Mozambique.**

| Variables | Model 1 | Model 2 |
|---|---|---|
| **Mother-level covariates** | | |
| BMI (kg/ m$^2$) = 18.50–24.99 | 1.780*** | 1.823*** |
| | (1.303–2.430) | (1.329–2.500) |
| BMI (kg/m m$^2$) = 25.00–29.99 | 2.144*** | 2.190*** |
| | (1.502–3.060) | (1.528–3.137) |
| BMI (kg/ m$^2$) > = 30 | 4.106*** | 4.268*** |
| | (2.453–6.870) | (2.537–7.179) |
| Maternal education (in single years) | 1.077*** | 1.079*** |
| | (1.052–1.102) | (1.054–1.105) |
| Age of the mother (in years) | 1.035*** | 1.039*** |
| | (1.018–1.052) | (1.022–1.056) |
| Employment status = working mother | 0.941 | 0.950 |
| | (0.813–1.088) | (0.821–1.098) |
| Parity | 0.943** | 0.942** |
| | (0.894–0.995) | (0.892–0.994) |
| Is Breastfeeding = Yes | 1.240*** | 1.070 |
| | (1.068–1.439) | (0.906–1.263) |
| Marital Status = Married | 0.977 | 1.009 |
| | (0.787–1.213) | (0.811–1.255) |
| Marital Status = Cohabiting | 1.278** | 1.289** |
| | (1.035–1.578) | (1.043–1.593) |
| Number of antenatal visits = 4+ visits | 1.055 | 0.942 |
| | (0.911–1.222) | (0.807–1.100) |
| **Household-level covariates** | | |
| Head of HH = Male | 0.874* | 0.887 |
| | (0.746–1.024) | (0.757–1.039) |
| Household size | 1.018 | 1.016 |
| | (0.988–1.048) | (0.986–1.046) |
| Number of children under 5 | 0.998 | 1.009 |
| | (0.908–1.098) | (0.916–1.111) |
| **Community-level covariate** | | |
| Urban residence = Urban | 1.185** | 1.181** |
| | (1.014–1.385) | (1.010–1.381) |
| **Child-level covariates** | | |
| Dietary Diversity (DD) > = 4 | | 0.888 |
| | | (0.752–1.048) |
| Age of the child (in months) | | 0.991*** |
| | | (0.986–0.995) |
| Gender = Male | | 0.931 |
| | | (0.811–1.069) |
| **Observations** | **5,711** | **5,711** |

95% Confidence Intervals (CIs) in parentheses; DD-Dietary diversity; HH-Household head; BMI-Body mass index

*** p<0.01

** p<0.05

* p<0.1

**Table 6. Adjusted odd ratios (aOR) of factors associated with normal growth among children living in better-off households, Nigeria.**

| Variables | Model 1 | Model 2 |
|---|---|---|
| **Mother-level covariates** | | |
| BMI (kg/ m$^2$) = 18.50–24.99 | 1.135 | 1.129 |
| | (0.934–1.379) | (0.931–1.371) |
| BMI (kg/ m$^2$) = 25.00–29.99 | 1.434*** | 1.435*** |
| | (1.157–1.777) | (1.159–1.777) |
| BMI (kg/ m$^2$) > = 30 | 1.697*** | 1.706*** |
| | (1.309–2.200) | (1.316–2.211) |
| Maternal education (in single years) | 1.067*** | 1.068*** |
| | (1.056–1.079) | (1.057–1.080) |
| Age of the mother (in years) | 1.033*** | 1.039*** |
| | (1.021–1.045) | (1.027–1.051) |
| Employment status = working mother | 1.072 | 1.106* |
| | (0.958–1.199) | (0.988–1.238) |
| Parity | 0.957*** | 0.954*** |
| | (0.925–0.989) | (0.922–0.986) |
| Is Breastfeeding = Yes | 1.537*** | 1.303*** |
| | (1.388–1.703) | (1.162–1.461) |
| Marital Status = Married | 0.806* | 0.803* |
| | (0.625–1.039) | (0.622–1.037) |
| Marital Status = Cohabiting | 0.866 | 0.846 |
| | (0.606–1.237) | (0.592–1.209) |
| Number of antenatal visits = 4+ visits | 1.080 | 0.877** |
| | (0.975–1.197) | (0.776–0.990) |
| **Household-level covariates** | | |
| Head of HH = Male | 0.850** | 0.856* |
| | (0.728–0.992) | (0.733–1.000) |
| Household size | 0.980** | 0.980** |
| | (0.963–0.998) | (0.962–0.998) |
| Number of children under 5 | 0.946* | 0.941** |
| | (0.893–1.002) | (0.888–0.997) |
| **Community-level covariate** | | |
| Urban residence = Urban | 1.004 | 1.013 |
| | (0.910–1.108) | (0.917–1.118) |
| **Child-level covariates** | | |
| Dietary Diversity (DD) > = 4 | | 1.055 |
| | | (0.904–1.231) |
| Age of the child (in months) | | 0.988*** |
| | | (0.984–0.991) |
| Gender = Male | | 0.785*** |
| | | (0.712–0.865) |
| **Observations** | **12,999** | **12,999** |

95% Confidence Intervals (CIs) in parentheses; DD-Dietary diversity; HH-Household head; BMI-Body mass index

*** p<0.01

** p<0.05

* p<0.1

## Discussion

This study investigated the maternal, child, household and community factors associated with normal (healthy) linear growth among children living in better-off households in DRC, Ghana, Kenya, Mozambique and Nigeria. The results highlighted the critical socio-demographic factors related to child growth outcomes and country-specific variations of these effects in the five countries. In the current analysis, higher maternal weight (measured by BMI) tends to have a significant positive impact on healthy growth among children living in better-off households in all countries except Ghana. Thus, maternal weight is a crucial determinant of positive child growth outcomes. Although a higher maternal weight has a positive effect on child healthy growth, interventions should target increasing maternal weight qualitatively for the benefit of both the mother and child. It is critical because of the negative effect of unhealthy weight on maternal health outcomes [51–53]. These findings are in line with the literature. Maternal nutrition has a significant positive impact on child linear growth in many settings [10–12]. In India, BMI had a substantial impact on child linear growth [11]. The preceding discussion illuminated the crucial role maternal nutrition plays in improving child nutritional status, although the pathways through which this happens may be complicated.

The vital role maternal education plays in promoting positive child health outcomes was observed in the present study. Our results showed that maternal years of schooling have significant positive effect on healthy linear growth among children in all the five countries except Kenya. Thus, suggesting that maternal education has the potential to promote the healthy growth of children living in better-off households. Interventions to improve child growth may have a positive impact on children living in these households. Previous studies documented the beneficial effects of maternal years of education on child growth outcomes. Improvement in maternal education was associated positively with a dramatic change in linear growth among pre-school children [10–12]. It may be the case because educated mothers tend to utilise both preventive and curative health care more [54, 55]. Educated mothers also tend to have more strongly committed attitude towards good childcare than uneducated mothers [56, 57]. Also, the more education the mother has, the more the likelihood that she is sensitive and responsive to caregiving duties [56, 57]. Also, there is evidence that children seemingly engage more positively with their mothers when maternal education is higher [12, 57]. All the above have positive effects on child growth outcomes. The literature discussed above, together with our study, demonstrated the importance of maternal education for positive child health outcomes. However, in Kenya, maternal education did not have a significant positive effect on child growth outcomes. This finding is puzzling as maternal education has consistently predict child growth in the literature. Further research is needed to disentangle the possible reasons accounting for the non-significant effect of maternal schooling on child growth outcomes in Kenya.

The benefits of breastfeeding to child health were illuminated in this study but only in two countries. Breastfeeding practice was associated significantly with the likelihood of healthy linear growth among children living in better-off households in Kenya and Nigeria. The finding may imply that mothers in these households should be encouraged to practice breastfeeding because of its beneficial effects on their children growth. These findings confirm the widely recognised benefits of breastfeeding for improved health and developmental outcomes [25–27]. On the contrary, breastfeeding showed a significant positive effect on child healthy linear growth in the models containing only the socio-demographic factors in Ghana, Mozambique and DRC. The statistically significant association disappeared after the inclusion of child-level covariates such as dietary diversity, age and sex in the final empirical models. Hence, whether breastfeeding will have a positive effect on healthy child growth in better-off households or not

is conditional on the inclusion or otherwise of child-level covariates. This finding corroborates previous research, which suggests the addition of child-level factors when evaluating the association of breastfeeding with anthropometric outcomes [57]. The non-significant positive effect of breastfeeding on child growth has previously been documented [58–60]. Thus, while breastfeeding is critical for positive child health outcomes, it is not always the case that its effects would be statistically significant.

Our analysis also illuminated negative determinants of healthy linear growth. Maternal work status was inversely related to healthy child growth in DRC. It implies that DRC mothers who are engaged in any form of work tend to have children who have poor linear growth relative to mothers who are not working. The negative effect may boil down to inadequate childcare due to limited time available to working mothers. A study in India concluded that a mother's employment compromises infant feeding and care, particularly so when mothers are not able to get alternative caregivers [61]. This study further reported that the compromises related to childcare and feeding outweigh the benefits from employment [61]. Other studies have shown that mothers working away from home spend less time with their children compared to mothers who are not working outside the home and therefore likely to have children who are undernourished [62, 63]. Although women who are working tend to have access to disposable income and consequently able to provide nutritious food for their children [13, 17, 64, 65], the above discussion showed that maternal employment, indeed, could negatively affect child growth outcomes. It is worthy to note that this analysis did not investigate the categories of work and their effect on child growth [12]. We are therefore unable to tell the independent impact of the various occupational groups on child linear growth. It is a limitation worth noting, as different occupations may have different effects.

Similarly, the results showed that higher maternal parity impact negatively on healthy linear growth in better-off households. The effect is most significant in Kenya (25%) followed by Ghana (23%), with the least impact being in Nigeria (5%). The findings suggest that higher parity has a stronger negative effect on child healthy growth in Kenya and Ghana relative to the other countries. The adverse effects of parity on child growth may be attributed to it being an essential factor that affects maternal depletion, particularly among high fertility mothers [66, 67]. Poor maternal health has the potential to compromise the mothers' ability to provide proper care for their children. The consequential effect of the lack of adequate care is poor child growth. Secondly, women with higher parity are likely to have many young children, who might compete for the available care resources, which can affect good care practices and consequently their children growth outcomes. Previous studies have documented that the higher the maternal parity, the less likely that their children will have positive growth outcomes [21, 22]. Children of multiparous mothers tend to have lower rates of growth and lower levels of childhood body mass index than children of nulliparous mothers [22]. The preceding discussion demonstrated that parity has a significant negative effect on child linear growth.

Our analysis also showed that a year increase in child's age was associated negatively with healthy linear growth in three (DRC, Mozambique and Nigeria) of the five countries. The implication is that older children living in better-off households are less likely to achieve healthy linear growth. These findings are consistent with previous research. Nshimyiryo and colleagues [68] observed that an increase in the child's age had a significant association with poor linear growth. For instance, children aged 6–23 months were at lower risk of poor growth than those in the older age group 24–59 months [68]. Also, being a male child is associated with less likelihood of healthy growth among children living in better-off households. Previous work has shown that poor linear growth was higher among male children as compared to female children [69]. Suggesting that male children tend to be more vulnerable to poor growth than their female counterparts in the same age group [70]. It might be due to preferences in

feeding practices or other types of exposures [70]. The findings could also be because boys are expected to grow at a slightly more rapid rate compared to girls, and their growth is perhaps more easily affected by nutritional deficiencies or other exposures [71].

## Strengths and limitations of the study

One significant advantage is the use of high quality, extensive nationally representative DHS data to investigate the factors associated with healthy linear growth among children living in better-off households. The comprehensive data make it possible for the findings to be generalised to the population of young children in the respective countries. The extensive data also help to produce more robust estimates of observed associations. The use of multi-country data unmasks differences and highlights commonalities in the effects of the correlates on child growth across countries. Revealing these differences may not have been possible with single country data.

Further, the height data used for computing the HAZ indicator were objectively measured, reducing possible misclassification. The novelty of this study is its focus on positive child growth outcomes rather than child growth deficiencies. A limitation worth mentioning is the cross-sectional nature of the data, which makes it challenging to disentangle potential reciprocal and otherwise complex causal relationships. We, therefore, restrict the interpretation of findings to mere associations between the explanatory variables and the outcome variables.

## Conclusions

Maternal weight (BMI) tends to have significant positive effects on healthy linear growth among children living in better-off households across all countries except Ghana. Interventions aimed at increasing maternal weight qualitatively are likely to be effective in improving the linear growth of children living in better-off households. Maternal years of education have significant positive effects on healthy linear growth in all the five countries except Kenya. Schooling has the potential to improve healthy linear growth among children in better-off households. Breastfeeding was associated with the likelihood of healthy linear growth in Kenya and Nigeria. Implying that mothers in better-off households should be encouraged to practice breastfeeding because of its beneficial effect on their children growth. Mothers work status is inversely related to healthy growth in DRC. Thus, in DRC, mothers who are engaged in any form of work tend to have children with poor growth relative to mothers who are not working. The results show that higher maternal parity associates negatively with healthy linear growth. The effect is most significant in Kenya (25%) followed by Ghana (23%), with the least impact being in Nigeria (5%). Thus, higher parity has a stronger negative impact on child healthy linear growth in Kenya and Ghana relative to the other countries.

## Acknowledgments

We wish to express our gratitude to The DHS Program, USA, for providing us access to the data. We also want to acknowledge the institutions of respective countries that played critical roles in the data collection processes.

## Author Contributions

**Conceptualization:** Dickson Abanimi Amugsi, Zacharie T. Dimbuene, Elizabeth W. Kimani-Murage.

**Data curation:** Dickson Abanimi Amugsi, Zacharie T. Dimbuene.

**Formal analysis:** Dickson Abanimi Amugsi, Zacharie T. Dimbuene, Elizabeth W. Kimani-Murage.

**Methodology:** Dickson Abanimi Amugsi, Zacharie T. Dimbuene, Elizabeth W. Kimani-Murage.

**Resources:** Elizabeth W. Kimani-Murage.

**Writing – original draft:** Dickson Abanimi Amugsi.

**Writing – review & editing:** Dickson Abanimi Amugsi, Zacharie T. Dimbuene, Elizabeth W. Kimani-Murage.

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
