## [Decision Letter · Decision Letter 0]

11 Nov 2019

PONE-D-19-27838

Positive and negative determinants of normal linear growth among pre-school children living in better-off households: an analysis of secondary data from sub-Saharan Africa

PLOS ONE

Dear Dr Amugsi,

Thank you for submitting your manuscript to PLOS ONE. After careful consideration, we feel that it has merit but does not fully meet PLOS ONE’s publication criteria as it currently stands. Therefore, we invite you to submit a revised version of the manuscript that addresses the points raised during the review process.

This is an interesting manuscript. Both well-written reviews of it note some merit in this manuscript, but have highlighted a number of areas where it needs improvement. I agree that it is very important to state why data from these 5 specific countries were used and why only data from children in higher income families were analysed. Both reviewers highlight the need to add some additional potential confounders to the statistical models, which I think is of key importance. I also think the influence of paternal factors on childhood linear growth merits further thought in a revised manuscript. Although these are the key points, all the limitations highlighted by the reviewers require dealing with in a revised manuscript.

We would appreciate receiving your revised manuscript by Dec 26 2019 11:59PM. To enhance the reproducibility of your results, we recommend that if applicable you deposit your laboratory protocols in protocols.io, where a protocol can be assigned its own identifier (DOI) such that it can be cited independently in the future. For instructions see: http://journals.plos.org/plosone/s/submission-guidelines#loc-laboratory-protocols

We look forward to receiving your revised manuscript.

Kind regards,

Clive J Petry, PhD

Academic Editor

PLOS ONE

Journal Requirements:

- Amugsi DA, Dimbuene ZT, Kimani-Murage EW, Mberu B, Ezeh AC. Differential effects of dietary diversity and maternal characteristics on linear growth of children aged 6-59 months in sub-Saharan  Africa: a multi-country analysis. Public health nutrition. 2017;20(6):1029-45

The text that needs to be addressed involves some paragraphs of the Introduction and of the Discussion.

In your revision ensure you cite all your sources (including your own works), and quote or rephrase any duplicated text outside the methods section. Further consideration is dependent on these concerns being addressed.

Additional Editor Comments (if provided):

As a very minor additional point I believe that there is typographical error in line 286 of the manuscript.

Reviewers' comments:

Reviewer's Responses to Questions

**Comments to the Author**

1. Is the manuscript technically sound, and do the data support the conclusions?

Reviewer #1: Partly

Reviewer #2: Partly

2. Has the statistical analysis been performed appropriately and rigorously? 

Reviewer #1: No

Reviewer #2: No

3. Have the authors made all data underlying the findings in their manuscript fully available?

Reviewer #1: Yes

Reviewer #2: Yes

4. Is the manuscript presented in an intelligible fashion and written in standard English?

Reviewer #1: Yes

Reviewer #2: No

5. Review Comments to the Author

Reviewer #1: Overall, you have presented an interesting manuscript of public health importance in sub-Saharan African countries. Reducing child growth faltering and ensuring optimal child growth remains a priority public health agenda among most developing countries in Africa and beyond. The study would serve as an additional input in nutrition policy makings. Your use of data from DHS as well as the inclusion of 5 countries increases the reliability and generalizability of the findings. Your analysis methodology, logistic regression, fits the objective of the study and the dichotomized nature of the outcome variable. The sections of the manuscript, introduction to conclusion, are presented in a clear and understandable way.

However, there are some major and minor issues that need to be addressed before the manuscript is accepted for publication.

Major comments (Methodology)

- Title: Consider restating the title. For example, “Factors associated with optimal growth status of under-5 children in….”

- Study population: It is not clear why you restricted the study to only the better-off households, given growth faltering is mainly the problem of the poor, not the rich. Please clarify it.

- In all DHS surveys, the samples are collected following a two-stage cluster sampling scheme. Besides, the DHS sampling scheme over-represents small regions (sub-national divisions). Thus, all analyses, descriptive as well as regression, should take into account the cluster design of the study and be done after weighting the data by the corresponding weighting variable (i.e. v005 in the DHS datasets). Regression outputs of the weighted data would be surely different from the regression outputs of unweighted data for the same measure. Otherwise, the estimates would be biased towards the over-represented states. At the current state, the manuscript doesn’t provide any information on how these issues were handled. Thus, I strongly recommend rerunning the data on the weighted sample and taking into account the design effect of the cluster sampling.

- Most of the variables with a proven influence on growth are not included in the analysis. For example, frequency of feeding, quality of complementary feeding, early initiation of breastfeeding, exclusive breastfeeding, vitamin A supplementation, history of infection, hygiene, sanitation, health care utilization, media exposure...etc. You included only dietary diversity variables. I am aware that these variables are included in the DHS datasets you used. What is your rationale to exclude these variables?

- The measurement of the predictor variables is not provided and need to be included during your revision. For example, I could not understand what the variable “breastfeeding’ refers to. It is not clear whether it refers to the duration of breastfeeding, exclusive breastfeeding or early initiation of breastfeeding. How you developed the dietary diversity scores is also not clear. I suggest including a separate section for ‘variables definition and measurement’.

- As far as I know DHS datasets, some of the variables you used are only available for children below 24 months of age. For example, the dataset doesn’t enable to develop dietary diversity score for those above 24 months. Given your samples are all under-5 children, it is not clear how you handle this issue. Please explain.

- A problem with DHS datasets is data missing. What was the level of data missing in your case? How you managed the data missing? Have you checked whether the missing cases were not different from the included cases? Please Explain. Additionally, it would be more informative to include a flow of chart of your ample selection process.

- You mentioned ‘sub-Saharan countries’ in the title but you used the data of only five countries. The selection of the five countries is not clear.

Minor comments

- Line 110-111: the statement ‘Statistical analyses that simultaneously examine the positive as well as negative determinants of normal child growth are missing’ is not right. There are various studies, including a study of mine, that examined the determinants of child growth simultaneously. Better to tone down the statement.

- Lines 188-189: You recoded middle, richer and richest households into “better-off households”. What is your rationale? Please provide a reference.

- Table 1: Check the heading for consistency (raw 2 in particular).

- Table 1: The SD columns are empty for the categorical variables, while SD values are provided for the variables on a continuous scale. Why not for the categorical variables? Besides, instead of SD, I would generally suggest presenting the 95%CI of the prevalence or means values of each variable.

- Lines 259: put space after 1.71 (aOR=1.75, 95% CI=1.17,2.62). Please check the whole manuscript for errors like this.

- Line 310: “The results shed new insights on both positive and negative determinants of child growth….” As your findings are not new for those in the field, please paraphrase the statement.

- Lines 325-326: How do you explain the unexpected finding that education was not significantly associated with growth in Kenya?

- Lines 342-343: Again how do you explain the null association between breastfeeding and growth in two of the five countries?

- Lines 422-423: “The novelty of this study is its focus on positive child growth outcomes”. This seems exaggerated given there are various studies that examined the determinants of child growth.

- There are editorial, punctuation and formatting errors. Example Line 76: the statement ‘Implementation of these programs more often than not places more emphasis on child…” Please check the manuscript thoroughly.

Reviewer #2: Abstract

Sentence 5: Higher maternal weight (measured by body mass index) � should it be maternal adiposity (?)

aOR is presented with CI, please also provide with p values

Introduction

Comments about language/writing style:

1. I would suggest improving the writing with more academic and indirect style (eg Line 65: “for children’s sake”, line 208: “we did this”, etc)

2. I would suggest consistency in the type of English pronunciation used, which I suppose should be American English (eg Line 76: programmes � programs)

Comments about content/analysis in general:

The paper addresses an important issue in public health by showing data on the child’s growth in LMIC with large number of subjects. While the topic is very exciting, LMIC portrays some complicated issues, relating to infections, poverty, lifestyle, and other morbidities.

1. There are some more factors that need to be considered in looking at children’s linear growth in LMIC. Checking at the DHS report, these data are available thus need to be included in the analyses: infections (eg malaria, TB, HIV both among children and parents), delivery method/assistance, parental tobacco smoking, sanitation, vaccination, and other morbidities (especially anaemia)

2. There have been more interests associating paternal factors and children’s growth, for example: https://www.sciencedirect.com/science/article/pii/S0021755716304375?via%3Dihub

Please consider paternal factors, otherwise, change the title to be maternal and early life nutritional determinant factors

3. The authors adjusted the models for child’s age and sex in the second and final model which has raised an important issue because: among all factors, gender has been established as the most factor defining an individual’s growth that growth standards separate the growth charts based on sex. I would suggest separating the analyses based on gender or use gender as a fixed factor in the model.

4. Using stunting (defined as HAZ<-2 at a time point) only as the predictor of linear growth is not wise because: 1) it does not differ stunting as growth faltering parameter with short stature, especially if not adjusted for parental height; 2) it does not reflect growth as a process, but only as a one-point measurement, 2) in paediatrics, the essential growth measurements include height, weight, adiposity (BMI as the crude/simplest measure, although is not ideal as body composition measure), and head circumference (only for children age 2 years or younger). Therefore, I would suggest complementing the analyses with catch-up/catch-down parameters (definition by ALSPAC study, BMJ 2000;320:967–71) or use height trajectories as dependent variable and then run linear mixed-models; and take into account weight-for-age, weight-for-age, and BMI-for-age scores.

5. The data analysis part of the methods section should explain more about the models being run, given the multiple understanding of “empirical regression models”. Is it regular regression with an emphasis on empirical use? Is it linear regression with empirical logit transformation? Is it logistic regression?

Comments about technical issue:

1. Table 1: Typos and misalignment, including column title of %/mean and SD applies for all countries but somehow missing for DRC and Ghana; “IS working” under working status ; Parity, Is breastfeeding, and number of antenatal visits should be typed in bold to ease reading the table

6. PLOS authors have the option to publish the peer review history of their article (what does this mean?). If published, this will include your full peer review and any attached files.

Reviewer #1: Yes: Shimels Hussien Mohammed

Reviewer #2: No

---

## [Author Response · Author response to Decision Letter 0]

30 Nov 2019

PONE-D-19-27838

Positive and negative determinants of normal linear growth among pre-school children living in better-off households: an analysis of secondary data from sub-Saharan Africa

PLOS ONE

Comments: This is an interesting manuscript. Both well-written reviews of it note some merit in this manuscript, but have highlighted a number of areas where it needs improvement. I agree that it is very important to state why data from these 5 specific countries were used and why only data from children in higher income families were analysed. Both reviewers highlight the need to add some additional potential confounders to the statistical models, which I think is of key importance. I also think the influence of paternal factors on childhood linear growth merits further thought in a revised manuscript. Although these are the key points, all the limitations highlighted by the reviewers require dealing with in a revised manuscript.

Response: We thank you for these positive comments. Please, find below our responses to the reviewers comments. Regarding why we used 5 countries, the following statement is captured in the manuscript “The selection of these five countries was informed by our previous work (12, 32)”

Journal Requirements:

Comment: 1. Please ensure that your manuscript meets PLOS ONE's style requirements, including those for file naming. The PLOS ONE style templates can be found at

Response: This now been done

Comment 2: We noticed you have some minor occurrence of overlapping text with the following previous publication(s), which needs to be addressed:

- Amugsi DA, Dimbuene ZT, Kimani-Murage EW, Mberu B, Ezeh AC. Differential effects of dietary diversity and maternal characteristics on linear growth of children aged 6-59 months in sub-Saharan Africa: a multi-country analysis. Public health nutrition. 2017;20(6):1029-45

The text that needs to be addressed involves some paragraphs of the Introduction and of the Discussion.

In your revision ensure you cite all your sources (including your own works), and quote or rephrase any duplicated text outside the methods section. Further consideration is dependent on these concerns being addressed.

Response: This has now been addressed

Comment 3: We suggest you thoroughly copyedit your manuscript for language usage, spelling, and grammar. If you do not know anyone who can help you do this, you may wish to consider employing a professional scientific editing service. 

Response: This has been done. The paper was carefully edited using Grammarly (Premium version). 

Reviewer #1: 

Comment: Overall, you have presented an interesting manuscript of public health importance in sub-Saharan African countries. Reducing child growth faltering and ensuring optimal child growth remains a priority public health agenda among most developing countries in Africa and beyond. The study would serve as an additional input in nutrition policy makings. Your use of data from DHS as well as the inclusion of 5 countries increases the reliability and generalizability of the findings. Your analysis methodology, logistic regression, fits the objective of the study and the dichotomized nature of the outcome variable. The sections of the manuscript, introduction to conclusion, are presented in a clear and understandable way.

However, there are some major and minor issues that need to be addressed before the manuscript is accepted for publication.

Response: We thank the reviewer for these positive comments

Major comments (Methodology)

Comment: Title: Consider restating the title. For example, “Factors associated with optimal growth status of under-5 children in….”

Response: The title now reads “Factors associated with normal linear growth among pre-school children living in better-off households: a multi-country analysis of nationally representative data”

Comment: Study population: It is not clear why you restricted the study to only the better-off households, given growth faltering is mainly the problem of the poor, not the rich. Please clarify it.

Response: The paper is not focusing on growth flattering (stunted children) but on children who are growing well (non-stunted). This sets this paper apart from the usual practices of focusing on children who are not growing well (growth flattering) and the risk and protective factors of the same. We limited the analysis to better-off households in order to elucidate the factors that make children in these households to have better health outcomes relative to children in poor households. This has already been explained in the manuscript.

Comment: In all DHS surveys, the samples are collected following a two-stage cluster sampling scheme. Besides, the DHS sampling scheme over-represents small regions (sub-national divisions). Thus, all analyses, descriptive as well as regression, should take into account the cluster design of the study and be done after weighting the data by the corresponding weighting variable (i.e. v005 in the DHS datasets). Regression outputs of the weighted data would be surely different from the regression outputs of unweighted data for the same measure. Otherwise, the estimates would be biased towards the over-represented states. At the current state, the manuscript doesn’t provide any information on how these issues were handled. Thus, I strongly recommend rerunning the data on the weighted sample and taking into account the design effect of the cluster sampling.

Response: This was done but not explicitly captured in the first version of the manuscript. We have now added this sentence to address the concern “Because the DHS utilised complex sample design, we accounted for design effect in all the analysis using parametres such as primary sampling unit, strata and weight”. 

Comment: Most of the variables with a proven influence on growth are not included in the analysis. For example, frequency of feeding, quality of complementary feeding, early initiation of breastfeeding, exclusive breastfeeding, vitamin A supplementation, history of infection, hygiene, sanitation, health care utilization, media exposure...etc. You included only dietary diversity variables. I am aware that these variables are included in the DHS datasets you used. What is your rationale to exclude these variables?

Response: We thank the reviewer for this comment. You are right, these variables are contained in the DHS data sets. However, to avoid making the results difficult to interpret and the tables unnecessary long (they are already long) or congested, after identifying the potential drivers (including the above referenced variables) of child growth per the literature and the UNICEF conceptual framework, we subjected them to bivariate analysis and included only statistical significant variables in the multivariable analysis. This is a standard practice in statistical analysis, since the researcher cannot include every potential factor in statistical models. Our own previous work and many others have shown that some the above referenced “proven” variables did not have a statistical significant relationship with child growth/health outcomes. 

Comment: The measurement of the predictor variables is not provided and need to be included during your revision. For example, I could not understand what the variable “breastfeeding’ refers to. It is not clear whether it refers to the duration of breastfeeding, exclusive breastfeeding or early initiation of breastfeeding. How you developed the dietary diversity scores is also not clear. I suggest including a separate section for ‘variables definition and measurement’.

Response: We thank the reviewer for this observation. We have now included a separate section in the manuscript titled “predictor variables”. See the paragraph below: 

We classified the predictor variables into child (dietary diversity, age and sex), maternal (body mass index, education, age, work status, parity, breastfeeding status, marital status, antenatal attendance), household (sex of household head, household size, number of children under 5 years) and community (place of residence) level variables. The child dietary diversity (DD) was created by counting the number of food groups the mother reported the child had consumed in the past 24h before the interview. In accordance with recommended procedures on the construction of the child DD indicator (42), we regrouped the food types in the data into seven (7) main categories (12): (i) grains, roots and tubers; (ii) legumes and nuts; (iii) dairy products; (iv) ﬂesh foods and organ meats; (v) vitamin A-rich fruits and vegetables; (vi) eggs; and (vii) other fruits and vegetables. A value of 1 was given for the child’s consumption of any of the food groups within 24h, while we assigned 0 for non-consumption (12). These scores were then summed up to obtain the DD score, ranging from 0 to 7, and used in the analysis as a categorical predictor variable. Also, a score of 1 was assigned to a mother who indicated she was still breastfeeding, and 0 for “no” response. Marital status was recoded into “not in union” “cohabiting” and “married”. Maternal body mass index (BMI), also referred to as Quetelet’s Index (43), was derived by dividing weight in kilograms by the squared height in meters (12). The BMI (kg/m2) was then classified into BMI<18.50 kg/m2 (underweight), BMI=18.50-24.99 kg/m2 (normal weight), BMI=25.0-29.9 kg/m2 (overweight) and BMI≥ 30.0 kg/m2 (obese) (44). We used BMI<18.50 kg/m2 (underweight) as a referenced category in the analysis. We selected the predictor variables based on the UNICEF conceptual framework of childcare (9) and the literature. We subjected the selected variables to bivariate analysis to establish their relationship with the outcome variable. Only statistically significant variables were included in the multivariable analysis.

Comment: As far as I know DHS datasets, some of the variables you used are only available for children below 24 months of age. For example, the dataset doesn’t enable to develop dietary diversity score for those above 24 months. Given your samples are all under-5 children, it is not clear how you handle this issue. Please explain.

Response: We thank the reviewer for the comment. This point is not entirely accurate. The DHS collects data on dietary intake of children up to 59 months old. These data are then used by analyst to construct the dietary diversity score. We have used this variable either as an outcome or predictor variable in not less than 15 scientific papers. 

Comment: A problem with DHS datasets is data missing. What was the level of data missing in your case? How you managed the data missing? Have you checked whether the missing cases were not different from the included cases? Please Explain. Additionally, it would be more informative to include a flow of chart of your ample selection process.

Response: This point is not also entirely accurate. Although the DHS data may have some issues with missing data, these issues are addressed before the data are made available to users as the following statement on their website suggests “One of the primary goals of The DHS Program is to produce high-quality data and make it available for analysis in a coherent and consistent form. However, national surveys in developing countries are prone to incomplete or partial reporting of responses. Additionally, complex questionnaires inevitably allow scope for inconsistent responses to be recorded for different questions. For the analyst this results in a data file containing incomplete or inconsistent data, complicating the analysis considerably. In order to avoid these problems, The DHS Program has adopted a policy of editing and imputation which results in a data file that accurately reflects the population studied and may be readily used for analysis. Primary data quality policies include”: (https://dhsprogram.com/data/Data-Quality-and-Use.cfm). 

We have been using the DHS data for several years and missing data have never been one of the issues we worry about. 

Comment: You mentioned ‘sub-Saharan countries’ in the title but you used the data of only five countries. The selection of the five countries is not clear.

Response: The title now reads “Factors associated with normal linear growth among pre-school children living in better-off households: a multi-country analysis of nationally representative data”

Minor comments

Comment: Line 110-111: the statement ‘Statistical analyses that simultaneously examine the positive as well as negative determinants of normal child growth are missing’ is not right. There are various studies, including a study of mine, that examined the determinants of child growth simultaneously. Better to tone down the statement.

Response: We have changed “missing” to limited. Nevertheless, it is important to point out that the analytical strategy used in this paper is somewhat different from what is common in the literature. Instead of focusing on growth flattering (stunting) and the associated risk and protective factors, we focused on normal child growth (non-stunted children) and the factors that explain why these children are not stunted. We will like to invite the reviewer (s) to look at the paper from this angle. 

Comment: Lines 188-189: You recoded middle, richer and richest households into “better-off households”. What is your rationale? Please provide a reference.

Response: We thank the reviewer for this comment. The literature on the positive health outcomes of children living in the middle to richest households is abound. What this study seeks to do is to understand the key factors that make children in these households to have better health outcomes relative to those living in poor households. 

Comment: Table 1: Check the heading for consistency (raw 2 in particular).

Response: This has been corrected

Comment: Table 1: The SD columns are empty for the categorical variables, while SD values are provided for the variables on a continuous scale. Why not for the categorical variables? Besides, instead of SD, I would generally suggest presenting the 95%CI of the prevalence or means values of each variable.

Response: We thank reviewer for this comment. We feel that it will not make scientific sense to present mean estimates for categorical variables. Also, Table 1 presents basic descriptive analysis, therefore, no confidence intervals are required. 

Comment: Lines 259: put space after 1.71 (aOR=1.75, 95% CI=1.17,2.62). Please check the whole manuscript for errors like this.

Response: This has been done

Comment: Line 310: “The results shed new insights on both positive and negative determinants of child growth….” As your findings are not new for those in the field, please paraphrase the statement.

Response: This concern has been addressed

Comment: Lines 325-326: How do you explain the unexpected finding that education was not significantly associated with growth in Kenya?

Response: We thank the reviewer for this observation. This is really not an unexpected findings. There are several studies (including ours), which did not show statistical significant effect of education on child health outcomes. Thus, it is not the case that education always has statistical significant effect on child health. 

Comment: Lines 342-343: Again how do you explain the null association between breastfeeding and growth in two of the five countries?

Response: We thank the reviewer for this comment. Breastfeeding is critical for positive child health outcomes, but it is not always the case that its effects would be statistically significant. The literature on the null association between breastfeeding and child growth outcomes is abound (we did show this in the manuscript). Indeed, there are several other studies (including ours) that showed inverse relationship between breastfeeding and child growth…the concept of reverse causality. 

Comment: Lines 422-423: “The novelty of this study is its focus on positive child growth outcomes”. This seems exaggerated given there are various studies that examined the determinants of child growth.

Response: We thank the reviewer for this comment. However, we respectfully disagree that this is an exaggeration. We have published several papers on childhood stunting (growth flattering) but this is the first time we are focusing on non-stunted children. This is also case in the literature. We will like to invite the reviewers to take note of the fact that in this paper, the outcome of interest is not stunted children but non-stunted. 

Comment: There are editorial, punctuation and formatting errors. Example Line 76: the statement ‘Implementation of these programs more often than not places more emphasis on child…” Please check the manuscript thoroughly.

Response: We thank the reviewer for this observation. The manuscript has been thoroughly checked using grammarly (premium version)

Reviewer #2: Abstract

Comment: Sentence 5: Higher maternal weight (measured by body mass index) ◊ should it be maternal adiposity (?)

Response: It is maternal weight (i.e. BMI)

Comment: aOR is presented with CI, please also provide with p values

Response: We thank the reviewer for this comment. We felt that presenting both confidence intervals and p-values will make the results difficult to read/congested. For statistical significance, the CIs are considered more robust than p-values, in our view. However, the level of statistical significance (p-values) has been presented at the end of every table (except Table 1, in which we did not test any statistical significance).

Introduction

Comments about language/writing style:

Comment: 1. I would suggest improving the writing with more academic and indirect style (eg Line 65: “for children’s sake”, line 208: “we did this”, etc)

Response: This has been corrected

Comment: 2. I would suggest consistency in the type of English pronunciation used, which I suppose should be American English (eg Line 76: programmes ◊ programs)

Response: We thank the reviewer for this comment. The manuscript is written in British English. We have carefully revised the manuscript using grammarly premium to ensure consistency.

Comments about content/analysis in general:

Comment: The paper addresses an important issue in public health by showing data on the child’s growth in LMIC with large number of subjects. While the topic is very exciting, LMIC portrays some complicated issues, relating to infections, poverty, lifestyle, and other morbidities.

Response: We thank the reviewer for this positive comment

comment: 1. There are some more factors that need to be considered in looking at children’s linear growth in LMIC. Checking at the DHS report, these data are available thus need to be included in the analyses: infections (eg malaria, TB, HIV both among children and parents), delivery method/assistance, parental tobacco smoking, sanitation, vaccination, and other morbidities (especially anaemia)

Response: We thank the reviewer for this comment. You are right, these variables are contained in the DHS data sets, although some of them may be country specific. However, to avoid making the results difficult to interpret and the tables unnecessary long (they are already long) or congested, after we’ve identified the potential drivers of child growth per the literature and the UNICEF conceptual framework, we subjected them to bivariate analysis and included only statistical significant variables in the multivariable analysis. This is a standard practice in statistical analysis, since the researcher cannot include every potential factor in statistical modelling.

Comment: 2. There have been more interests associating paternal factors and children’s growth, for example: https://www.sciencedirect.com/science/article/pii/S0021755716304375?via%3Dihub

Please consider paternal factors, otherwise, change the title to be maternal and early life nutritional determinant factors

Response: The title now reads “Factors associated with normal linear growth among pre-school children living in better-off households: a multi-country analysis of nationally representative data”

Comment: 3. The authors adjusted the models for child’s age and sex in the second and final model which has raised an important issue because: among all factors, gender has been established as the most factor defining an individual’s growth that growth standards separate the growth charts based on sex. I would suggest separating the analyses based on gender or use gender as a fixed factor in the model.

Response: Adjusting for gender in the multivariable models addresses this issue. We do not think it is necessary to conduct separate analysis, using gender as a stratification variable. 

Comment: 4. Using stunting (defined as HAZ<-2 at a time point) only as the predictor of linear growth is not wise because: 1) it does not differ stunting as growth faltering parameter with short stature, especially if not adjusted for parental height; 2) it does not reflect growth as a process, but only as a one-point measurement, 2) in paediatrics, the essential growth measurements include height, weight, adiposity (BMI as the crude/simplest measure, although is not ideal as body composition measure), and head circumference (only for children age 2 years or younger). Therefore, I would suggest complementing the analyses with catch-up/catch-down parameters (definition by ALSPAC study, BMJ 2000;320:967–71) or use height trajectories as dependent variable and then run linear mixed-models; and take into account weight-for-age, weight-for-age, and BMI-for-age scores.

Response: We thank the reviewer for this comment. It is not clear what the reviewer is inviting us to do. However, we will like to point out that this paper is not about STUNTED children but NON-STUNTED. Regarding the issue of growth as a process, these are cross sectional data, therefore, examining growth as a process may be a difficult thing to do. This is would have been possible with longitudinal data.

Comment: 5. The data analysis part of the methods section should explain more about the models being run, given the multiple understanding of “empirical regression models”. Is it regular regression with an emphasis on empirical use? Is it linear regression with empirical logit transformation? Is it logistic regression?

Response: We thank the reviewer for this comment. We stated in the manuscript that we employed logistic regression in the analysis.

Comments about technical issue:

Comment: 1. Table 1: Typos and misalignment, including column title of %/mean and SD applies for all countries but somehow missing for DRC and Ghana; “IS working” under working status ; Parity, Is breastfeeding, and number of antenatal visits should be typed in bold to ease reading the table

Response: This has been done

---

## [Decision Letter · Decision Letter 1]

18 Dec 2019

PONE-D-19-27838R1

Factors associated with normal linear growth among pre-school children living in better-off households: a multi-country analysis of nationally representative data

PLOS ONE

Dear Dr Amugsi,

Thank you for submitting your manuscript to PLOS ONE. After careful consideration, we feel that it has merit but does not fully meet PLOS ONE’s publication criteria as it currently stands. Therefore, we invite you to submit a revised version of the manuscript that addresses the points raised during the review process.

The revised version of this manuscript only seems to have made a relatively small amount of progress. The manuscript still clearly has merit, but I agree with both reviewers that their concerns have not been adequately addressed. I think that the quality of the manuscript merits the authors being given one more chance, but also that the authors need to more carefully consider each of the points restated by the reviewers in a second revision.

We would appreciate receiving your revised manuscript by Feb 01 2020 11:59PM. To enhance the reproducibility of your results, we recommend that if applicable you deposit your laboratory protocols in protocols.io, where a protocol can be assigned its own identifier (DOI) such that it can be cited independently in the future. For instructions see: http://journals.plos.org/plosone/s/submission-guidelines#loc-laboratory-protocols

We look forward to receiving your revised manuscript.

Kind regards,

Clive J Petry, PhD

Academic Editor

PLOS ONE

Reviewers' comments:

Reviewer's Responses to Questions

**Comments to the Author**

1. If the authors have adequately addressed your comments raised in a previous round of review and you feel that this manuscript is now acceptable for publication, you may indicate that here to bypass the “Comments to the Author” section, enter your conflict of interest statement in the “Confidential to Editor” section, and submit your "Accept" recommendation.

Reviewer #1: (No Response)

Reviewer #2: (No Response)

2. Is the manuscript technically sound, and do the data support the conclusions?

Reviewer #1: Partly

Reviewer #2: Yes

3. Has the statistical analysis been performed appropriately and rigorously? 

Reviewer #1: No

Reviewer #2: No

4. Have the authors made all data underlying the findings in their manuscript fully available?

Reviewer #1: Yes

Reviewer #2: Yes

5. Is the manuscript presented in an intelligible fashion and written in standard English?

Reviewer #1: Yes

Reviewer #2: No

6. Review Comments to the Author

Reviewer #1: Thank you for addressing some of my previous comments. Variables definitions have been included and editorial issues have been thoroughly addressed. However, I’m sorry to indicate that most of my comments remain unaddressed, particularly the methodological issues. Besides, your rebuttals are largely unsatisfactory and unsupported by relevant references. It would have been better if had referenced your responses, instead of justifying just from the point of the authors’ personal experience and expertise in the field. Let the work speak for itself. At this stage, I do not recommend accepting the manuscript as it would affect the literature pool as well as the reputation of the journal. Please find my comments below.

In my previous comment, I recommended the importance of stating the criteria for the selection of 5 countries, out of the over 50 African countries. Just for example, why Kenya is included but not Ethiopia? Your response has been “The selection of these five countries was informed by our previous work (12, 32)”. I checked these references (12, 32), both of which don’t provide any information on why data from these 5 specific countries were used. Thus, the comment still remains an important issue worthy of addressing.

Line 203: You stated “Only statistically significant variables were included in the multivariable analysis”. This is wrong and a major methodological bias of the work. Selection of variables for inclusion in multivariable models is done at relaxed p-values (often at P 0.20 or 0.25); such that, variables that demonstrated P<0.20 or P<0.25 would be included in the multivariable model. Statistical significance at P<0.05 is appropriate only at the final model. Besides, I would like to put at your note that variables could also be included in multivariable models based on their theoretical (biological) relevance. Please refer to the following and other references on models building:

- Bursac Z, Gauss CH, Williams DK, Hosmer DW. Purposeful selection of variables in logistic regression. Source Code Biol Med. 2008;3:17. Published 2008 Dec 16. doi:10.1186/1751-0473-3-17

Earlier, I recommended including the following variables in your analysis: frequency of feeding, quality of complementary feeding, early initiation of breastfeeding, exclusive breastfeeding, vitamin A supplementation, history of infection, hygiene, sanitation, health care utilization, media exposure...etc. However, your response goes “After identifying the potential drivers (including the above referenced variables) of child growth per the literature and the UNICEF conceptual framework, we subjected them to bivariate analysis and included only statistical significant variables in the multivariable analysis. This is a standard practice in statistical analysis, since the researcher cannot include every potential factor in statistical models. Our own previous work and many others have shown that some the above referenced “proven” variables did not have a statistical significant relationship with child growth/health outcomes.

Your response above is not satisfactory for many reasons including: a) I would rather argue that the evidence in support of the importance of these variables for optimal child growth and health is more robust than the evidence against their importance. Please check meta-analysis reports on the impact of caregivers’ educational status on child health and nutritional outcomes. A lack of association in your and some others’ data doesn’t necessarily mean that the variables are of no importance in reality. There are many reasons for lack of association between two variables, particularly when using cross-sectional collected data, b) you included < 10 predictor variables, which is really not much given child growth is a multifactorial condition. Most of your variables are even sociodemographic ones like maternal age, household size, number of children in the household, which we refer to ‘distal factors’. The proximal factors, immediate determinants, are largely unaccounted. Thus, it is still better to re-run the analysis by including these variables because as far as I know, the datasets enable to do that. If you couldn’t do that for logistic reasons like table size, you need to state it as a limitation, c) There is no mention of the bi-variable analysis in the statistical method section. You also did not present any result from the bi-variable analysis. This leads me to doubt whether you have done it or not. If you have done it, please state in the method the specific tests done for the different scales.

Line 80-82. “The present study fills this gap by providing robust evidence on the critical factors associated with healthy…..” Please tone down the statement as this study doesn’t take into account many of the important child growth and nutrition enhancing variables, like meal frequency, vaccination, hygiene, micronutrient supplementation, etc.

I am still not satisfied with your response to my comment on how you developed the dietary diversity score for children above 24 months. To the best of my knowledge, the infant and young child feeding practice (IYCFP) indicators guideline you followed to develop the diversity scores is aimed to be used for only under-24 months’ children. Which guideline or methodological reference is justified your use of the variable for those above 24 months? Besides, the dietary data collected by DHS for under-24 and above-24months are quite different. That is clearly stated in all questionnaires, reports and guidelines produced by DHS itself. For more, please check the following references:

- The 2018 Guide to DHS Statistics: for dietary diversity (please type 410 at your PDF page locator)

https://www.dhsprogram.com/pubs/pdf/DHSG1/Guide_to_DHS_Statistics_DHS-7.pdf

- For IYCF indicators: please check the following guidelines by WHO. 2007. Indicators for assessing infant and young child feeding practices.

Part I Definitions: http://www.who.int/nutrition/publications/infantfeeding/9789241596664/en/

Part II Measurement: http://www.who.int/nutrition/publications/infantfeeding/9789241599290/en/

- A sample Nigeria DHS 2013 report: on page 211 (on pdf viewer) or page 184 (inside the document) “The 2013 NDHS used a 24-hour recall method to collect data on infant and young child feeding for all last-born children under age 2 living with their mothers. Table 11.3……”

https://dhsprogram.com/publications/publication-fr293-dhs-final-reports.cfm

Your response, “…We have been using the DHS data for several years and missing data have never been one of the issues we worry about”, is also wrong. Data missing is actually one of the challenges in all DHS datasets. Cognizant of that, even reports of the DHS agency itself always include statements about missing information. You could randomly check any DHS report of any country. The tables and reports include a separate category labelled “missing”. I put here the Nigeria DHS 2013 report for your reference on this issue. Just search ‘Missing’ on your pdf and see the results. I even have checked some of the datasets and learned the data is not complete on all variables as you said. Thus, I advise you to address the issue, state the level of missing as well as whether the missing cases were significantly different from the included ones. I think it is not difficult to present at least the level of missing.

https://dhsprogram.com/publications/publication-fr293-dhs-final-reports.cfm

Your response, “We feel that it will not make scientific sense to present mean estimates for categorical variables”, doesn’t make sense statistically. Mean estimates, as well as standard deviations, can be presented for both categorical and non-categorical variables. Sex is a categorical variable. You can state mean like this the mean age of females is XX year (SD), the proportion of females is XX% (SD)…. The comment refers to table 1. Thus, either include SD for all variables or remove it entirely. Alternatively, instead of SD, you could present the point estimates all variables with their corresponding 95%CI. Whichever way you do, be consistent.

You stated, in your rebuttal, that the lack of association between education status of caregivers and the growth of children is not something unexpected. However, I would still argue that it is an unusual finding, given the known benefits of better education in improving child health and nutritional status. Thus, I still recommend you to state some of the potential reasons for the lack of the association between educational status and child growth in the Kenyan data.

Minor comments

Please also check the document thoroughly for editorials errors like following.

Line 115: “using healthy growth as the primaryr outcome variable….” primary

Reviewer #2: In general, I have observed the improved quality of the paper and I think the content of this paper is important, especially for policy-making in the LMIC setting. However, I would encourage the authors to make further revisions, as below:

1. Writing style:

I appreciate that there have been some improvements in the text, but I’m not sure that using the premium service of Grammarly (i.e. automatic software checking) is the best/most sufficient option for academic writing like this. I would suggest asking a native speaker from your group to thoroughly proofread the manuscript (checking the grammar accuracy, using consistently academic diction in the whole text, etc) or hiring a professional service. I’ve highlighted a few examples of mistakes that still exist in this revised manuscript:

a. Line 115: “primaryr” (typo)

b. Line 124: “The objective of this study, therefore, is…” (should be “was”)

c. Line 333: “The results highlight….” (should be “highlighted”)

d. Line 383-384: “Maternal work status is….” (should be “was”; “maternal work status” in some places is inconsistently written as “maternal working status”, might read better as “maternal employment status”?)

e. Text in general: many grammar inconsistencies. You may find this link helpful: https://www.nature.com/scitable/topicpage/effective-writing-13815989/

2. Line 31, line 279, etc: Maternal weight (measured by BMI)

This may be too meticulous, but I still don’t get it why you measure weight by BMI parameter? I might suggest writing this as “weight” (as in not taking height into account) OR adiposity (measured by BMI)

3. I appreciate that you have changed the title to address the reviewers’ comments, but it still needs editing to address these issues: 1) after all, this study is based on African setting so it needs to mention in the title, 2) factors taken into account in the analyses were limited to maternal factors (not considering paternal influence) and infant baseline characteristics (sex and feeding, not including morbidities e.g. infections, anaemia, etc.)

4. Data analysis:

Line 245-253: As sex and age are the two most robust and determinant factors in postnatal growth, I am still not sure why you put both factors in the second covariates group, rather than being the fixed factor or in the first covariates group?

5. Although the paper is focused more on non-stunted children, growth and infections/morbidities are inseparable in LMIC setting, so please add some justification of excluding of those factors in your analyses

6. Please pay attention to the tables and check them carefully:

a. Table 1: please use the same font size throughout; move “child-level covariates” one level down (not in the same level as %/mean and SD); “sex of child” replace with “gender” (no need to mention “child” because it’s under the child-level covariates); “IS working” should be written as “Is working”; separate “parity” and “is breastfeeding” with space so they wouldn’t seem to be part of “working status” category; etc; please check again the whole content carefully

b. Table 2-6: I would suggest writing the names of variables more clearly; for example, “working status = is working” to be “working mother”, “Sex of child = Male” to be “Male sex”, etc.

7. PLOS authors have the option to publish the peer review history of their article (what does this mean?). If published, this will include your full peer review and any attached files.

Reviewer #1: Yes: Shimels Hussien Mohammed

Reviewer #2: No

---

## [Author Response · Author response to Decision Letter 1]

4 Feb 2020

PONE-D-19-27838R1

Factors associated with normal linear growth among pre-school children living in better-off households: a multi-country analysis of nationally representative data

PLOS ONE

Reviewer #1: 

COMMENT: Thank you for addressing some of my previous comments. Variables definitions have been included and editorial issues have been thoroughly addressed. However, I’m sorry to indicate that most of my comments remain unaddressed, particularly the methodological issues. Besides, your rebuttals are largely unsatisfactory and unsupported by relevant references. It would have been better if had referenced your responses, instead of justifying just from the point of the authors’ personal experience and expertise in the field. Let the work speak for itself. At this stage, I do not recommend accepting the manuscript as it would affect the literature pool as well as the reputation of the journal. Please find my comments below.

RESPONSE: We thank the reviewer for this comment. We will ensure that every response is referenced if necessary. We are not sure about the reviewer’s assertion regarding the literature pool and the reputation of the Journal. As far as we know, this analysis, which focuses on exogenous drivers (socio-demographic factors) of child growth will contribute significantly to the literature considering the key exogenous variables (16) used in the analysis, and the analytical strategy…focusing on resources for health instead of risk factors, and positive health outcomes instead of negative (disease etc). 

COMMENT: In my previous comment, I recommended the importance of stating the criteria for the selection of 5 countries, out of the over 50 African countries. Just for example, why Kenya is included but not Ethiopia? Your response has been “The selection of these five countries was informed by our previous work (12, 32)”. I checked these references (12, 32), both of which don’t provide any information on why data from these 5 specific countries were used. Thus, the comment still remains an important issue worthy of addressing.

RESPONSE: We thank the reviewer for this comment. We revised the sentence as follows; “The selection of the five countries was based on our previous analysis using the same countries and data (12, 32)” It is difficult to give any scientific justification for the use of the five countries and not the 50+ in Africa and over 100 across the low and middle income countries. The decision of the number of countries to use in an analysis is usually at the discretion of the researcher/analyst. Some analysts use 1 country, others 2 or more countries etc. Please refer to a few examples below. https://bmjopen.bmj.com/content/6/9/e012615/ 10.1017/s0021932017000505/https://doi.org/10.1136/bmjopen-2017-017344/
https://doi.org/10.1017/s1368980016003426/
https://doi.org/10.1371/journal.pone.0136748 / https://www.ncbi.nlm.nih.gov/pmc/articles/PMC4409817/ etc.

COMMENT: Line 203: You stated “Only statistically significant variables were included in the multivariable analysis”. This is wrong and a major methodological bias of the work. Selection of variables for inclusion in multivariable models is done at relaxed p-values (often at P 0.20 or 0.25); such that, variables that demonstrated P<0.20 or P<0.25 would be included in the multivariable model. Statistical significance at P<0.05 is appropriate only at the final model. Besides, I would like to put at your note that variables could also be included in multivariable models based on their theoretical (biological) relevance. Please refer to the following and other references on models building:

- Bursac Z, Gauss CH, Williams DK, Hosmer DW. Purposeful selection of variables in logistic regression. Source Code Biol Med. 2008;3:17. Published 2008 Dec 16. doi:10.1186/1751-0473-3-17

RESPONSE: We thank the reviewer for this comment. We agree with the reviewer that biologically important variables may not have to necessarily be statistical significant at the bivariate analysis stage before they are included in the multivariable analysis. And we always endeavour to acknowledge this in our publications. Refer to examples below. We have added this sentence to the manuscript “variables that were not statistically significant but considered biologically important were included in the multivariable analysis”. The type of analysis suggested in the above referenced paper are beyond the scope of our study.

https://bmjopen.bmj.com/content/6/9/e012615 /https://www.nature.com/articles/s41598-018-26991-4

We however, humbly disagree with the reviewer assertion that non-significant variables should be included, whether they are considered critical by the analyst or not. This will create uninterpretable parameters and unmanageable tables. This is supported by the paper the reviewer shared: “Some methodologists suggest inclusion of all clinical and other relevant variables in the model regardless of their significance in order to control for confounding. This approach, however, can lead to numerically unstable estimates and large standard errors” 

There are tens of potential drivers of child health. Since it is not possible to include all of them, there must be inclusion and exclusion criteria, and that is the purpose of the bivariate analysis…this is what is mostly done in the literature. Regarding the biases in the selection of the variables, this arises when you arbitrarily select variables without subjecting them to scientific test. In the bivariate analysis, every identified potential predictor is given the chance to be included in the multivariable analysis and the excluded variables are usually those that did not meet the inclusion criteria. 

COMMENT: Earlier, I recommended including the following variables in your analysis: frequency of feeding, quality of complementary feeding, early initiation of breastfeeding, exclusive breastfeeding, vitamin A supplementation, history of infection, hygiene, sanitation, health care utilization, media exposure...etc. However, your response goes “After identifying the potential drivers (including the above referenced variables) of child growth per the literature and the UNICEF conceptual framework, we subjected them to bivariate analysis and included only statistical significant variables in the multivariable analysis. This is a standard practice in statistical analysis, since the researcher cannot include every potential factor in statistical models. Our own previous work and many others have shown that some the above referenced “proven” variables did not have a statistical significant relationship with child growth/health outcomes.

Your response above is not satisfactory for many reasons including: a) I would rather argue that the evidence in support of the importance of these variables for optimal child growth and health is more robust than the evidence against their importance. Please check meta-analysis reports on the impact of caregivers’ educational status on child health and nutritional outcomes. A lack of association in your and some others’ data doesn’t necessarily mean that the variables are of no importance in reality. There are many reasons for lack of association between two variables, particularly when using cross-sectional collected data, b) you included < 10 predictor variables, which is really not much given child growth is a multifactorial condition. Most of your variables are even sociodemographic ones like maternal age, household size, number of children in the household, which we refer to ‘distal factors’. The proximal factors, immediate determinants, are largely unaccounted. Thus, it is still better to re-run the analysis by including these variables because as far as I know, the datasets enable to do that. If you couldn’t do that for logistic reasons like table size, you need to state it as a limitation, c) There is no mention of the bi-variable analysis in the statistical method section. You also did not present any result from the bi-variable analysis. This leads me to doubt whether you have done it or not. If you have done it, please state in the method the specific tests done for the different scales.

RESPONSE: We thank the reviewer for this comment. Although we subject several variables to bivariate analysis as indicated in our early response, the focus of the paper is on exogenous variables (socio-demographic factors). After the bivariate analysis, we settled on 16 predictor variables (and not <10 variables). The reason for deciding to focus on only the exogenous variables (unfortunately we did bring this up in our earlier response) is captured in the paper as follows: “In this analysis, we included only the basic/exogenous factors (socio-demographic factors) in our empirical models. We did this because there is evidence that in examining the association between child growth outcomes and exogenous factors, the proximate factors (endogenous factors) are usually excluded to avoid biased estimation of the regression parameters of the exogenous factors (47-49). It is the case because the proximate factors are pathways through which the exogenous factors influence child nutrition (48)”

Regarding the bivariate analysis, we included this sentence to address the reviewer’s concern: “ We tested statistical significant associations of the categorical variables using chi-square, while continuous variables were tested using t-test” 

COMMENT: Line 80-82. “The present study fills this gap by providing robust evidence on the critical factors associated with healthy…..” Please tone down the statement as this study doesn’t take into account many of the important child growth and nutrition enhancing variables, like meal frequency, vaccination, hygiene, micronutrient supplementation, etc.

RESPONSE: The sentence has been revised.

COMMENT: I am still not satisfied with your response to my comment on how you developed the dietary diversity score for children above 24 months. To the best of my knowledge, the infant and young child feeding practice (IYCFP) indicators guideline you followed to develop the diversity scores is aimed to be used for only under-24 months’ children. Which guideline or methodological reference is justified your use of the variable for those above 24 months? Besides, the dietary data collected by DHS for under-24 and above-24months are quite different. That is clearly stated in all questionnaires, reports and guidelines produced by DHS itself. For more, please check the following references:

- The 2018 Guide to DHS Statistics: for dietary diversity (please type 410 at your PDF page locator)

https://www.dhsprogram.com/pubs/pdf/DHSG1/Guide_to_DHS_Statistics_DHS-7.pdf

- For IYCF indicators: please check the following guidelines by WHO. 2007. Indicators for assessing infant and young child feeding practices.

Part I Definitions: http://www.who.int/nutrition/publications/infantfeeding/9789241596664/en/

Part II Measurement: http://www.who.int/nutrition/publications/infantfeeding/9789241599290/en/

- A sample Nigeria DHS 2013 report: on page 211 (on pdf viewer) or page 184 (inside the document) “The 2013 NDHS used a 24-hour recall method to collect data on infant and young child feeding for all last-born children under age 2 living with their mothers. Table 11.3……”

https://dhsprogram.com/publications/publication-fr293-dhs-final-reports.cfm

RESPONSE: We thank the reviewer for this comment. In the DHS final data there are two sets of dietary variables: Children dietary and mothers dietary data respectively. The child dietary variables are not disaggregated by age. Also, the dietary diversity indicator used for the IYCF is the same indicator recommended for use in children under five years. The construction of the dietary diversity score in our study was based strictly on the recommended procedures of creating the same. Below are few examples of our studies and others that used this indicator for children under 5 years. 

https://www.ncbi.nlm.nih.gov/pmc/articles/PMC6044523/#pone.0200235.ref013/
https://doi.org/10.1017/s1368980016003426/
https://doi.org/10.1017/s1368980016003426/
https://doi.org/10.1136/bmjopen-2014-005340/
https://doi.org/10.1017/
https://doi.org/10.1371/journal.pone.0136748

COMMENT: Your response, “…We have been using the DHS data for several years and missing data have never been one of the issues we worry about”, is also wrong. Data missing is actually one of the challenges in all DHS datasets. Cognizant of that, even reports of the DHS agency itself always include statements about missing information. You could randomly check any DHS report of any country. The tables and reports include a separate category labelled “missing”. I put here the Nigeria DHS 2013 report for your reference on this issue. Just search ‘Missing’ on your pdf and see the results. I even have checked some of the datasets and learned the data is not complete on all variables as you said. Thus, I advise you to address the issue, state the level of missing as well as whether the missing cases were significantly different from the included ones. I think it is not difficult to present at least the level of missing.

https://dhsprogram.com/publications/publication-fr293-dhs-final-reports.cfm

RESPONSE: We thank the reviewer for this comment. We agree with reviewer that DHS included column on missing data in their recent reports. However, missing data is not a major issue in their reports. For example, the Nigerian DHS report has an average missingness of <1% (i.e. all tables combine). This is the case with other reports. In our analysis, we did not observe missingness as a major issue, therefore we did not see the need to conduct missing data analysis e.g. explicit imputation, since it has already been taken care of by the DHS during the preparation of the data files (see statement below from the DHS). Indeed, missing data analysis is almost non-existence in the literature using the DHS data. 

“One of the primary goals of The DHS Program is to produce high-quality data and make it available for analysis in a coherent and consistent form. However, national surveys in developing countries are prone to incomplete or partial reporting of responses. Additionally, complex questionnaires inevitably allow scope for inconsistent responses to be recorded for different questions. For the analyst this results in a data file containing incomplete or inconsistent data, complicating the analysis considerably. In order to avoid these problems, The DHS Program has adopted a policy of editing and imputation which results in a data file that accurately reflects the population studied and may be readily used for analysis. Primary data quality policies include”: (https://dhsprogram.com/data/Data-Quality-and-Use.cfm)”

COMMENT: Your response, “We feel that it will not make scientific sense to present mean estimates for categorical variables”, doesn’t make sense statistically. Mean estimates, as well as standard deviations, can be presented for both categorical and non-categorical variables. Sex is a categorical variable. You can state mean like this the mean age of females is XX year (SD), the proportion of females is XX% (SD)…. The comment refers to table 1. Thus, either include SD for all variables or remove it entirely. Alternatively, instead of SD, you could present the point estimates all variables with their corresponding 95%CI. Whichever way you do, be consistent. 

RESPONSE: We thank the reviewer for this comment. We still insist that statistically, it does not make sense to estimate means and standard deviations for categorical variables. The Editor and the statisticians at PLOS ONE may also give their opinion this. We do not also think it is necessary to present point estimates and CIs when describing the characteristics of samples. It is usually a basic analysis to give a picture of the sample before the main analysis and should, therefore, not be an issue for scientific interrogation. 

COMMENT: You stated, in your rebuttal, that the lack of association between education status of caregivers and the growth of children is not something unexpected. However, I would still argue that it is an unusual finding, given the known benefits of better education in improving child health and nutritional status. Thus, I still recommend you to state some of the potential reasons for the lack of the association between educational status and child growth in the Kenyan data.

RESPONSE: We thank the reviewer for this comment. This sentences have been added to address the reviewer’s concern: “However, in Kenya, maternal education did not have a significant effect on child growth outcomes. This finding is puzzling as maternal education has consistently predict child growth in the literature. Further research is needed to disentangle the possible reasons accounting for the non-significant effect of maternal schooling on child growth outcomes in Kenya” 

Minor comments

COMMENT: Please also check the document thoroughly for editorials errors like following.

Line 115: “using healthy growth as the primaryr outcome variable….” Primary

RESPONSE: We thank the reviewer for this comment. This error has now been corrected

Reviewer #2: 

COMMENT: In general, I have observed the improved quality of the paper and I think the content of this paper is important, especially for policy-making in the LMIC setting. However, I would encourage the authors to make further revisions, as below:

RESPONSE: We thank the reviewer for this positive comment.

1. Writing style:

COMMENT: I appreciate that there have been some improvements in the text, but I’m not sure that using the premium service of Grammarly (i.e. automatic software checking) is the best/most sufficient option for academic writing like this. I would suggest asking a native speaker from your group to thoroughly proofread the manuscript (checking the grammar accuracy, using consistently academic diction in the whole text, etc) or hiring a professional service. I’ve highlighted a few examples of mistakes that still exist in this revised manuscript:

RESPONSE: We thank the reviewer for this comment. The manuscript has been thoroughly proofread.

COMMENT: a. Line 115: “primaryr” (typo)

RESPONSE: This has now been corrected

COMMENT: b. Line 124: “The objective of this study, therefore, is…” (should be “was”)

RESPONSE: This has now been corrected

COMMENT: c. Line 333: “The results highlight….” (should be “highlighted”)

RESPONSE: Has been corrected. 

COMMENT: d. Line 383-384: “Maternal work status is….” (should be “was”; “maternal work status” in some places is inconsistently written as “maternal working status”, might read better as “maternal employment status”?)

RESPONSE: This has now been corrected

COMMENT: e. Text in general: many grammar inconsistencies. You may find this link helpful: https://www.nature.com/scitable/topicpage/effective-writing-13815989/

RESPONSE: We thank the reviewer for providing this link. It is helpful.

COMMENT: 2. Line 31, line 279, etc: Maternal weight (measured by BMI)

This may be too meticulous, but I still don’t get it why you measure weight by BMI parameter? I might suggest writing this as “weight” (as in not taking height into account) OR adiposity (measured by BMI)

RESPONSE: We thank the reviewer for this comment. We appreciate the reviewer’s concern. However, we used weight in this context to describe the level of the BMI. e.g. BMI <18.5 kg/m^2 = is described as “underweight”; BMI =18.5-24.99 “normal weight” and so on. You cannot tell whether an individual is underweight, overweight or obese by using the absolute value from a weighing scale. You have to compute the BMI (weight in kg divided by height in m^2) before you are able to tell the weight status of the individual. This is the context within which we used the term “weight” 

COMMENT: 3. I appreciate that you have changed the title to address the reviewers’ comments, but it still needs editing to address these issues: 1) after all, this study is based on African setting so it needs to mention in the title, 2) factors taken into account in the analyses were limited to maternal factors (not considering paternal influence) and infant baseline characteristics (sex and feeding, not including morbidities e.g. infections, anaemia, etc.)

RESPONSE: We Thank the reviewer for this comment. We have changed the title to read “Socio-demographic factors associated with normal linear growth among pre-school children living in better-off households: a multi-country analysis of nationally representative data” 

The paternal variables such as education and employment status were not significant in the bivariate analysis. We therefore excluded them and others to make the tables manageable. 

Just to add that the paper focuses of exogenous variables (socio-demographic factors). The following is a text that explains why we use this approach. “In this analysis, we included only the basic/exogenous factors (socio-demographic factors) in our empirical models. We did this because there is evidence that in examining the association between child growth outcomes and exogenous factors, the proximate factors (endogenous factors) are usually excluded to avoid biased estimation of the regression parameters of the exogenous factors (47-49). It is the case because the proximate factors are pathways through which the exogenous factors influence child nutrition (48).

4. Data analysis:

COMMENT: Line 245-253: As sex and age are the two most robust and determinant factors in postnatal growth, I am still not sure why you put both factors in the second covariates group, rather than being the fixed factor or in the first covariates group?

RESPONSE: We thank the reviewer for this comment. The sex and age are not candidate predictor variables but control variables in the this analysis. As explained above, this analysis focused more on exogenous variables. We therefore feel that controlling for them in the model should be enough. 

COMMENT: 5. Although the paper is focused more on non-stunted children, growth and infections/morbidities are inseparable in LMIC setting, so please add some justification of excluding of those factors in your analyses

RESPONSE: We thank the reviewer for this comment. The following is the justification for the variables used in the analysis: “In this analysis, we included only the basic/exogenous factors (socio-demographic factors) in our empirical models. We did this because there is evidence that in examining the association between child growth outcomes and exogenous factors, the proximate factors (endogenous factors) are usually excluded to avoid biased estimation of the regression parameters of the exogenous factors (47-49). It is the case because the proximate factors are pathways through which the exogenous factors influence child nutrition (48)” Further, we also used the bivariate results to decide on which variables to include in the multivariable analysis. This decision was taken to ensure the stability of the models and make the results tables manageable

COMMENT: 6. Please pay attention to the tables and check them carefully:

a. Table 1: please use the same font size throughout; move “child-level covariates” one level down (not in the same level as %/mean and SD); “sex of child” replace with “gender” (no need to mention “child” because it’s under the child-level covariates); “IS working” should be written as “Is working”; separate “parity” and “is breastfeeding” with space so they wouldn’t seem to be part of “working status” category; etc; please check again the whole content carefully

b. Table 2-6: I would suggest writing the names of variables more clearly; for example, “working status = is working” to be “working mother”, “Sex of child = Male” to be “Male sex”, etc.

RESPONSE: This has now been corrected.

---

## [Decision Letter · Decision Letter 2]

20 Feb 2020

PONE-D-19-27838R2

Socio-demographic factors associated with normal linear growth among pre-school children living in better-off households: a multi-country analysis of nationally representative data

PLOS ONE

Dear Dr Amugsi,

Thank you for submitting your manuscript to PLOS ONE. After careful consideration, we feel that it has merit but does not fully meet PLOS ONE’s publication criteria as it currently stands. Therefore, we invite you to submit a revised version of the manuscript that addresses the points raised during the review process.

This version of the manuscript is a definite improvement on the last one. Both reviewers still have slight concerns about the manuscript, however, and have suggested further revisions, albeit that these are more minor than before. I think that the most important comment to address is the one about statistical analysis from reviewer 2. The remaining comments from both reviewers are minor. Please note that reviewer 2 has given two options for dealing their concern about the statistical analysis, so even this should be easy to achieve to a satisfactory standard. I look forward to receiving one further revised version.

We would appreciate receiving your revised manuscript by Apr 05 2020 11:59PM. To enhance the reproducibility of your results, we recommend that if applicable you deposit your laboratory protocols in protocols.io, where a protocol can be assigned its own identifier (DOI) such that it can be cited independently in the future. For instructions see: http://journals.plos.org/plosone/s/submission-guidelines#loc-laboratory-protocols

We look forward to receiving your revised manuscript.

Kind regards,

Clive J Petry, PhD

Academic Editor

PLOS ONE

Reviewers' comments:

Reviewer's Responses to Questions

**Comments to the Author**

1. If the authors have adequately addressed your comments raised in a previous round of review and you feel that this manuscript is now acceptable for publication, you may indicate that here to bypass the “Comments to the Author” section, enter your conflict of interest statement in the “Confidential to Editor” section, and submit your "Accept" recommendation.

Reviewer #1: (No Response)

Reviewer #2: All comments have been addressed

2. Is the manuscript technically sound, and do the data support the conclusions?

Reviewer #1: Yes

Reviewer #2: Yes

3. Has the statistical analysis been performed appropriately and rigorously? 

Reviewer #1: Yes

Reviewer #2: No

4. Have the authors made all data underlying the findings in their manuscript fully available?

Reviewer #1: Yes

Reviewer #2: Yes

5. Is the manuscript presented in an intelligible fashion and written in standard English?

Reviewer #1: Yes

Reviewer #2: Yes

6. Review Comments to the Author

Reviewer #1: I noticed a substantial improvement in the manuscript. Thank you for addressing my comments duly and I feel the manuscript is now close to acceptance for publication after addressing the following minor comments.

- Line 42: Please add a comma after Nigeria in the sentence ‘….Nigeria respectively”

- Line 42: Please correct the figures in the sentence “A child being a male was associated with 16% (aOR=0.82, 95% CI=0.68, 0.98)” Is it 18% or 16%. From the ORs, it seems 18% (=100%– 82%) would be the right figure. Besides, please check and correct the figures at line 309.

- Line 74: Please define ‘WHO and UNICEF’. Abbreviations should be defined at their first appearance. Please check the manuscript thoroughly for other abbreviations, too.

- Line 132: “LIMCs” is not right. Please correct it by ‘LMICs”

- Lines 142-145: As your study population are under-5 children, wouldn’t it be better to state ‘all under-5 children in the selected households were eligible for inclusion in the study’? I mean instead of stating about adults.

- Line 151: add after mother ‘or caregivers’ for caregivers also provided information when mothers weren’t available.

- Line 364: predict or predicted?

- Please check the manuscript thoroughly for other typos, punctuation and grammar issues.

- Good luck with your modification.

Reviewer #2: I can observe that the authors clearly consider all inputs given and have attempted to improve the manuscript. However, with all due respect, I still think these aspects should be addressed before the manuscript can be published.

1. Statistical analyses

I understand the point you’re coming from to not include all factors suggested in your model by emphasising that you focus more on the exogenous factors, but I still think that the basic endogenous/inherent factors are still inseparable and need to be included in the models (e.g. this article: https://doi.org/10.1017/S1368980016002640). If you still disagree, please can you add this when stating your limitation?

2. Table

a. Table 1 still contains different font size

b. Table 2 and 3 look good now

c. Table 4-6 got double colons in the title

7. PLOS authors have the option to publish the peer review history of their article (what does this mean?). If published, this will include your full peer review and any attached files.

Reviewer #1: Yes: Shimels Hussien Mohammed

Reviewer #2: No

---

## [Author Response · Author response to Decision Letter 2]

24 Feb 2020

Reviewer #1: 

COMMENT: I noticed a substantial improvement in the manuscript. Thank you for addressing my comments duly and I feel the manuscript is now close to acceptance for publication after addressing the following minor comments.

RESPONSE: We thank the reviewer for this positive comment.

COMMENT: - Line 42: Please add a comma after Nigeria in the sentence ‘….Nigeria respectively”

RESPONSE: This has been done

COMMENT: - Line 42: Please correct the figures in the sentence “A child being a male was associated with 16% (aOR=0.82, 95% CI=0.68, 0.98)” Is it 18% or 16%. From the ORs, it seems 18% (=100%– 82%) would be the right figure. Besides, please check and correct the figures at line 309.

RESPONSE: This has been done

COMMENT: - Line 74: Please define ‘WHO and UNICEF’. Abbreviations should be defined at their first appearance. Please check the manuscript thoroughly for other abbreviations, too.

RESPONSE: This has now been done

COMMENT: - Line 132: “LIMCs” is not right. Please correct it by ‘LMICs”

RESPONSE: This has been corrected

COMMENT: - Lines 142-145: As your study population are under-5 children, wouldn’t it be better to state ‘all under-5 children in the selected households were eligible for inclusion in the study’? I mean instead of stating about adults.

RESPONSE: The description in lines 142-145 refers to DHS methodology and necessarily that of the present analysis.

COMMENT:- Line 151: add after mother ‘or caregivers’ for caregivers also provided information when mothers weren’t available.

RESPONSE: This has been done

COMMENT: - Line 364: predict or predicted?

RESPONSE: Predict

COMMENT: - Please check the manuscript thoroughly for other typos, punctuation and grammar issues.

- Good luck with your modification.

RESPONSE: Thank you

Reviewer #2: 

COMMENT: I can observe that the authors clearly consider all inputs given and have attempted to improve the manuscript. However, with all due respect, I still think these aspects should be addressed before the manuscript can be published.

RESPONSE: Thank you for this positive comment

COMMENT: 1. Statistical analyses

I understand the point you’re coming from to not include all factors suggested in your model by emphasising that you focus more on the exogenous factors, but I still think that the basic endogenous/inherent factors are still inseparable and need to be included in the models (e.g. this article: https://doi.org/10.1017/S1368980016002640). If you still disagree, please can you add this when stating your limitation?

RESPONSE: We thank the reviewer for this comment. We feel that the paragraph below (as captured in the manuscript) strongly justify why we focus on exogenous variables. Suffice to say that focusing on exogenous variables does not mean that the endogenous variables are ignored but that they are considered as pathways through which exogenous variables affect child growth. For example, maternal education which has found variously as a strong predictor of child health outcome does not have a direct effect on child health. Its effect is through improved caring practices, better use health services etc. by educated mothers. In light of the forgoing, it may not be necessary to indicate non-inclusion of some endogenous variables as a limitation of the analysis.

“However, the extended UNICEF model of care guided this analysis (9, 46). This framework suggests that child survival, growth and development are influenced by a web of factors, with three underlying determinants being food security, healthcare and a healthy environment, and care for children and women (9). Basic determinants have a direct influence on these underlying determinants. These basic determinants may be described as “exogenous” determinants, which influence child nutrition through their effect on the intervening proximate determinants (underlying determinants). The underlying factors are, therefore, endogenously determined by the exogenous factors (46). In this analysis, we included only the basic factors (socio-demographic factors) in our empirical models. We did this because there is evidence that in examining the association between child growth outcomes and exogenous factors, the proximate factors (endogenous factors) are usually excluded to avoid biased estimation of the regression parameters of the exogenous factors (47-49). It is the case because the proximate factors are pathways through which the exogenous factors influence child nutrition (48).”

COMMENTS:

2. Table

a. Table 1 still contains different font size

b. Table 2 and 3 look good now

c. Table 4-6 got double colons in the title

RESPONSE: These errors have now been corrected

---

## [Editor Report · Decision Letter 3]

26 Feb 2020

Socio-demographic factors associated with normal linear growth among pre-school children living in better-off households: a multi-country analysis of nationally representative data

PONE-D-19-27838R3

Dear Dr. Amugsi,

We are pleased to inform you that your manuscript has been judged scientifically suitable for publication and will be formally accepted for publication once it complies with all outstanding technical requirements.

With kind regards,

Clive J Petry, PhD

Academic Editor

PLOS ONE

Additional Editor Comments (optional):

I have judged this manuscript to be suitable for publication. However I think that the manuscript would have been better if the authors had included comment about the lack of use of endogenous factors in their statistical models as a study limitation, as suggested by one of the reviewers.
---

## [Editor Report · Acceptance letter]

28 Feb 2020

PONE-D-19-27838R3 

Socio-demographic factors associated with normal linear growth among pre-school children living in better-off households: a multi-country analysis of nationally representative data 

Dear Dr. Amugsi:

I am pleased to inform you that your manuscript has been deemed suitable for publication in PLOS ONE. Congratulations! Your manuscript is now with our production department. 

With kind regards,

on behalf of

Dr. Clive J Petry 

Academic Editor

PLOS ONE